# Sensitization of neonatal rat lumbar motoneuron by the inflammatory pain mediator bradykinin

Mouloud Bouhadfane[1], Attila Kaszás[2], Balázs Rózsa[3,4], Ronald M Harris-Warrick[5], Laurent Vinay[1], Frédéric Brocard[1]*

[1]Institut de Neurosciences de la Timone (UMR7289), Aix-Marseille Université and CNRS, Marseille, France; [2]Institut de Neuroscience des Systèmes (UMR1106), Aix Marseille Université and INSERM, Marseille, France; [3]Two-Photon Imaging Center, Institute of Experimental Medicine, Hungarian Academy of Sciences, Budapest, Hungary; [4]Faculty of Information Technology and Bionics, Pázmány Péter Catholic University, Budapest, Hungary; [5]Department of Neurobiology and Behavior, Cornell University, Ithaca, United States

**Abstract** Bradykinin (Bk) is a potent inflammatory mediator that causes hyperalgesia. The action of Bk on the sensory system is well documented but its effects on motoneurons, the final pathway of the motor system, are unknown. By a combination of patch-clamp recordings and two-photon calcium imaging, we found that Bk strongly sensitizes spinal motoneurons. Sensitization was characterized by an increased ability to generate self-sustained spiking in response to excitatory inputs. Our pharmacological study described a dual ionic mechanism to sensitize motoneurons, including inhibition of a barium-sensitive resting $K^+$ conductance and activation of a nonselective cationic conductance primarily mediated by $Na^+$. Examination of the upstream signaling pathways provided evidence for postsynaptic activation of $B_2$ receptors, G protein activation of phospholipase C, InsP3 synthesis, and calmodulin activation. This study questions the influence of motoneurons in the assessment of hyperalgesia since the withdrawal motor reflex is commonly used as a surrogate pain model.

*For correspondence: frederic.
brocard@univ-amu.fr

## Introduction

The nanopeptide bradykinin (Bk) is an important mediator of pain and inflammation (*Dray and Perkins, 1993*; *Calixto et al., 2000*). It causes hyperalgesia (*Manning et al., 1991*; *Dalmolin et al., 2007*) by exciting and/or sensitizing components of the pain pathway, including primary afferent terminals, sensory ganglia, and dorsal horn neurons (*Dray and Perkins, 1988*; *Dray et al., 1988*; *Thayer et al., 1988*; *McGuirk and Dolphin, 1992*; *Rueff and Dray, 1993*; *Jeftinija, 1994*; *Cesare et al., 1999*; *Wang et al., 2005*). Consistent with this, systemic blockade of Bk receptors produces analgesia (*Steranka et al., 1988*; *Correa and Calixto, 1993*; *Perkins et al., 1993*; *Levy and Zochodne, 2000*; *Ferreira et al., 2002*).

Although much has been reported on the action of Bk on the sensory system, its effects on motoneurons remain under-explored. Earlier and indirect evidence suggested that Bk may transynaptically activate motoneurons through activation of transmitter release from primary nociceptive afferents (*Dunn and Rang, 1990*). However, kinin, cleaved to Bk by kallikrein in inflammatory responses, is extensively distributed throughout the motor areas of the CNS (*Walker et al., 1995*; *Raidoo and Bhoola, 1998*) and in particular in the ventral horn of the spinal cord (*Lopes and Couture, 1997*; *Li et al., 1999*). Furthermore, Bk receptors have been detected in

**eLife digest** When we accidentally place our hand on a hot stove, we normally experience a painful sensation that starts with the sensory nerves under our skin. These nerves respond by transmitting electrical impulses to our brain, where the painful sensation is then processed. At the same time, these impulses are also transmitted to the motor nerves that control the muscles in our hand to trigger an immediate reflex to withdraw the hand from the hot stove. Pain therefore has a useful role as it can reduce how bad an injury is.

People with a condition called hyperalgesia have an increased sensitivity to pain. This condition can result from a chemical called bradykinin 'sensitizing' the sensory nerves, causing them to transmit more electrical impulses in response to pain than normal. This makes the injury feel much more painful, and can make the pain last for longer than is beneficial.

It was less clear whether bradykinin also affects motor nerves and so triggers a withdrawal reflex. By recording the electrical activity of motor nerve cells taken from the spinal cords of newborn rats, Bouhadfane et al. now show that these motor nerves become more active when exposed to bradykinin.

Nerve cells generate electrical signals when ions—such as potassium, sodium, and calcium ions—move through channels in the membranes of the cell. Therefore, to investigate how bradykinin influences the electrical activity of motor nerves, Bouhadfane et al. exposed the cells to drugs that inhibit particular ion channels. This revealed that bradykinin sensitizes the motor nerves by blocking a type of potassium ion channel and activating another ion channel that mainly transports sodium ions. Furthermore, Bouhadfane et al. were able to identify the signaling pathways that allow bradykinin to affect the motor nerve cells.

The study implies that the neuronal circuitry for pain does not rely exclusively on sensory nerve cells but should also integrate motor nerve cells. A future challenge remains in developing a protocol to resolve the contribution of motor nerve cells to hyperalgesia assessed by reflex withdrawal.

the membrane of spinal motoneurons (*Lopes et al., 1995*) suggesting that pain-related behaviors such as the withdrawal reflex may also arise from a direct Bk-evoked sensitization of motoneurons. The purpose of our study was to determine whether direct activation of Bk receptors sensitizes lumbar motoneurons of neonatal rats and, if so, to characterize the ionic mechanisms involved.

## Results

### Bk potentiates the gain of sensory inputs into the motor system

From an in vitro hemicord preparation, the direct application of Bk (4–8 µM) above the ventral horn column (L3–L5, *Figure 1A*) evoked sustained ventral root activity without inducing antidromic discharge from the dorsal roots (*Figure 1B*, *left*). This motor output was not exclusively caused by local reverberating circuits, because it persisted in the presence of the glutamate receptor antagonist kynurenate at a concentration that blocked the monosynaptic reflex (1.5 mM, *Figure 1B*, *right* and *Figure 1C*). We tested whether a change in the motoneuronal processing of sensory inputs by Bk might contribute to an increase in the motor output. To avoid sustained ventral root discharge, subthreshold concentrations of Bk (0.5–1 µM) were used. With these low concentrations of Bk, the motoneuron spiking probability in response to supramaximal dorsal root stimulation was increased (*Figure 1D–F*). Specifically, Bk had no effect on the transient short latency reflexes (number of events: $144 \pm 16$ for control vs $143 \pm 14$ during Bk; $p = 0.68$, Wilcoxon paired test, n = 7) but recruited a long-lasting reflex (number of events: $537 \pm 104$ for control vs $2493 \pm 1207$ during Bk; $p = 0.015$, Wilcoxon paired test, n = 7) such that the distribution of peristimulus time histograms (PSTHs) shifted from unimodal to bimodal (*Figure 1E,F*). This long-lasting reflex has been previously shown to result from sustained firing of motoneurons involved in muscle spasms (*Bennett et al., 2004*; *Li et al., 2004*).

### Bk increases excitability of lumbar motoneurons and promotes self-sustained spiking

To explore a potential sensitization of the firing properties of lumbar motoneurons by Bk, we recorded motoneurons in whole cell configuration from in vitro slice preparations (*Table 1* and

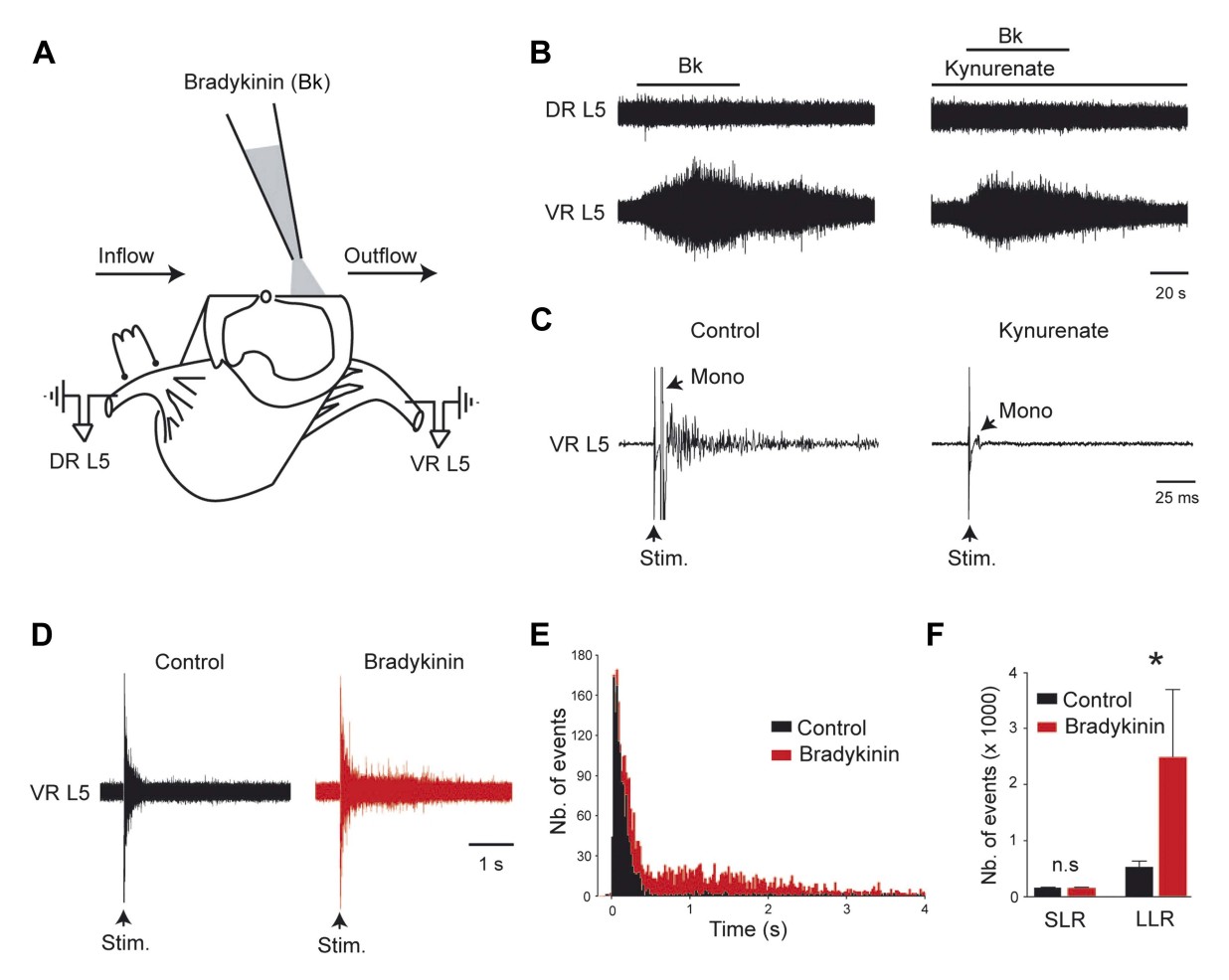

Figure 1. Bradykinin potentiates the gain of sensory inputs into the motor system. (A) Drawing of a midsagittally hemisected rat spinal cord illustrating localized Bk application to the lumbar motor column, and dorsal (DR) and ventral (VR) roots used for reflex testing. (B) Responses to ventral application of bradykinin (Bk, 4 μM pipette concentration) recorded via the lumbar L5 dorsal (DR L5) and ventral (VR L5) roots before (left) and after (right) the fast glutamatergic synaptic transmission was blocked by kynurenate (1.5 mM). (C) Ventral root response to ipsilateral dorsal root stimulation before (left) and after (right) application of kynurenate (1.5 mM). Single arrows indicate the monosynaptic reflex (mono) and the stimulus artifact (stim). (D) Five superimposed responses recorded from an L5 ventral root induced by stimulations of the ipsilateral dorsal root before (control, black trace) and after local application of low concentrations of Bk (1 μM pipette concentration, red trace). (E) Average peristimulus time histogram (PSTH, bin width: 20 ms) of L5 ventral root recordings collected before (black) and after (red) the local application of Bk (1 μM pipette concentration). (F) Group means quantification of the PSTH for the transient short latency and long-lasting reflexes computed over a window 10–40 ms and 500–4000 ms post stimulus, respectively, before (black) and after (red) local application of Bk. Error bars indicate SEM. *$p < 0.05$ (Wilcoxon paired test).

The following figure supplement is available for figure 1:

**Figure supplement 1**. At the right, five superimposed responses recorded under spantide (2 μM) from the L5 ventral root of an hemichord preparation and induced by stimulations of the ipsilateral dorsal root before (control, black trace) and after local application of low concentrations of Bk (1 μM pipette concentration, red trace).

Figure 2). From a resting potential adjusted to −70 mV with bias current, an incrementing series of hyperpolarizing and depolarizing pulses was delivered before and after bath application of Bk (4–8 μM). A few minutes (2–3 min) after the application of Bk, the major effect was an increase in motoneuron excitability, as reflected by a lower rheobase (528 ± 118 pA for control vs 299 ± 156 pA during Bk; $p = 0.028$, Wilcoxon paired test, n = 10 cells; Table 1 and Figure 2A,B). As a consequence, Bk evoked a leftward shift of the f-I curve to lower current values and a slight increase in the slope of the f-I curve (0.05 ± 0.004 Hz/pA for control vs 0.06 ± 0.006 Hz/pA during Bk; $p = 0.027$, Wilcoxon

**Table 1**. Effects of bradykinin on passive and active membrane properties of lumbar motoneurons

|  | Control | Bradykinin |
|---|---|---|
| N | 10 | 10 |
| Rm (MΩ) | 55.6 ± 5.4* | 62.9 ± 7.2 |
| AP amp (mV) | 67.0 ± 2.3* | 63.7 ± 2.2 |
| AP dur (ms) | 0.51 ± 0.02 | 0.51 ± 0.02 |
| AP threshold (mV) | −50.0 ± 2.1† | −52.8 ± 2.5 |
| f-I slope (Hz/pA) | 0.05 ± 0.004* | 0.06 ± 0.006 |
| Rheobase (pA) | 528 ± 118* | 299 ± 156 |
| sADP (mV) | 11.2 ± 1.4† | 18 ± 2.6 |
| AHP amp (mV) | −9.1 ± 0.9* | −11.2 ± 1.5 |
| AHP dur (ms) | 41.1 ± 2.5* | 48.6 ± 3.4 |
| Sag (%) | 12 ± 2.6 | 12.3 ± 2.5 |

Statistical significance was assessed by a Wilcoxon paired test.
*p < 0.05,
†p < 0.01,
n = number of cells. Mean firing frequency was measured at two times the rheobase.
AHP = afterhyperpolarization.

paired test, n = 10 cells; *Table 1* and *Figure 2B*). Thus, current steps that elicited only a single spike prior to Bk typically induced repetitive spiking during Bk application (*Figure 2A*). The higher motoneuron excitability was associated with an apparent increase in input resistance (55.6 ± 5.4 MΩ for control vs 62.9 ± 7.2 MΩ during Bk; p = 0.004, Wilcoxon paired test, n = 10 cells; *Table 1*) as seen by the increased slope of the voltage responses to hyperpolarizing pulses (*Figure 2A,C*). Bk lowered the threshold for action potential generation (−50.0 ± 2.1 mV for the control vs −52.8 ± 2.5 mV during Bk; p = 0.009, Wilcoxon paired test, n = 10 cells; *Table 1*) which, although not wider (0.51 ± 0.02 ms for control vs 0.51 ± 0.02 ms during Bk; p = 0.77, Wilcoxon paired test, n = 10 cells; *Table 1*), was smaller in amplitude (67.0 ± 2.3 mV for control vs 63.6 ± 2.2 mV during Bk; p = 0.049, Wilcoxon paired test, n = 10 cells; *Table 1* and *Figure 2D*). Hyperpolarizing current steps evoked a marked depolarizing sag (inward rectification) which is often associated with activation of the hyperpolarization-activated inward current $I_h$ (*Figure 2A*); the amplitude of the sag was not affected by Bk (12 ± 2.6% for control vs 12.3 ± 2.5% during Bk; p = 0.95, Wilcoxon paired test, n = 10 cells; *Table 1*). However, the medium afterhyperpolarization (mAHP) was enhanced both in amplitude and duration (amplitude: −9.1 ± 0.9 mV for the control vs −11.2 ± 1.5 mV for Bk, p = 0.02; duration: 41.1 ± 2.5 ms for control vs 48.6 ± 3.4 ms for Bk, p = 0.014; Wilcoxon paired test, n = 10 cells; *Table 1*; *Figure 2E*). In a recent publication, we demonstrated that at temperatures above 30°C, neonatal rat lumbar motoneurons show marked bistability, characterized by their ability to generate a slow afterdepolarization (sADP) that outlasted a brief high amplitude depolarizing pulse (*Bouhadfane et al., 2013*). Bk increased the amplitude of the post-stimulus sADP (11.2 ± 1.4 mV for control vs 18 ± 2.6 mV for Bk; p = 0.008, Wilcoxon paired test, n = 10 cells; *Table 1*) and triggered self-sustained spiking in all motoneurons tested (*Figure 2F*). These results show that Bk sensitizes motoneurons by increasing their excitability and promoting bistability.

## Bk depolarizes lumbar motoneurons through a likely direct postsynaptic activation

To test whether Bk directly excites motoneurons, we recorded activity from 200 lumbar motoneurons in response to a brief (≤2 min) bath application of Bk (8 μM). Among them, 154 (77%) displayed a reversible membrane depolarization (*Figure 3A*) while the remaining 46 had no responses. The onset of the depolarization was slow, taking close to 1 min to peak, and subsided slowly over 3–5 min during washout. During the peak of the depolarizing response, spikelets, indicative of electrical coupling to other neurons, were sometimes seen (*Figure 3A*, *inset*). The amplitude of the depolarization was not changed after blockade of glutamatergic input with kynurenate (1.5 mM) or by CNQX (10 μM) and AP-5 (20 μM) (92.9 ± 17.8% of the control; p = 0.62, Wilcoxon paired test, n = 4 cells; *Figure 2A*, *right*). These effects were thus not exclusively caused by a feed-forward network of excitatory synaptic input to motoneurons, but rather probably involved a direct postsynaptic action. Supporting this idea, the Bk-induced depolarization persisted in the presence of 0.5–1 μM TTX (12.9 ± 0.6 mV; n = 154; *Figure 3B*). The effects of Bk were fully reversible without tachyphylaxis; when Bk was repetitively applied at intervals of 20 min, the amplitude of the second response was not significantly reduced from the first response (95.6 ± 13.9% of the first response; p = 0.22, Wilcoxon paired test, n = 7 cells; *Figure 3B*). Previous indirect evidence suggested that Bk's actions

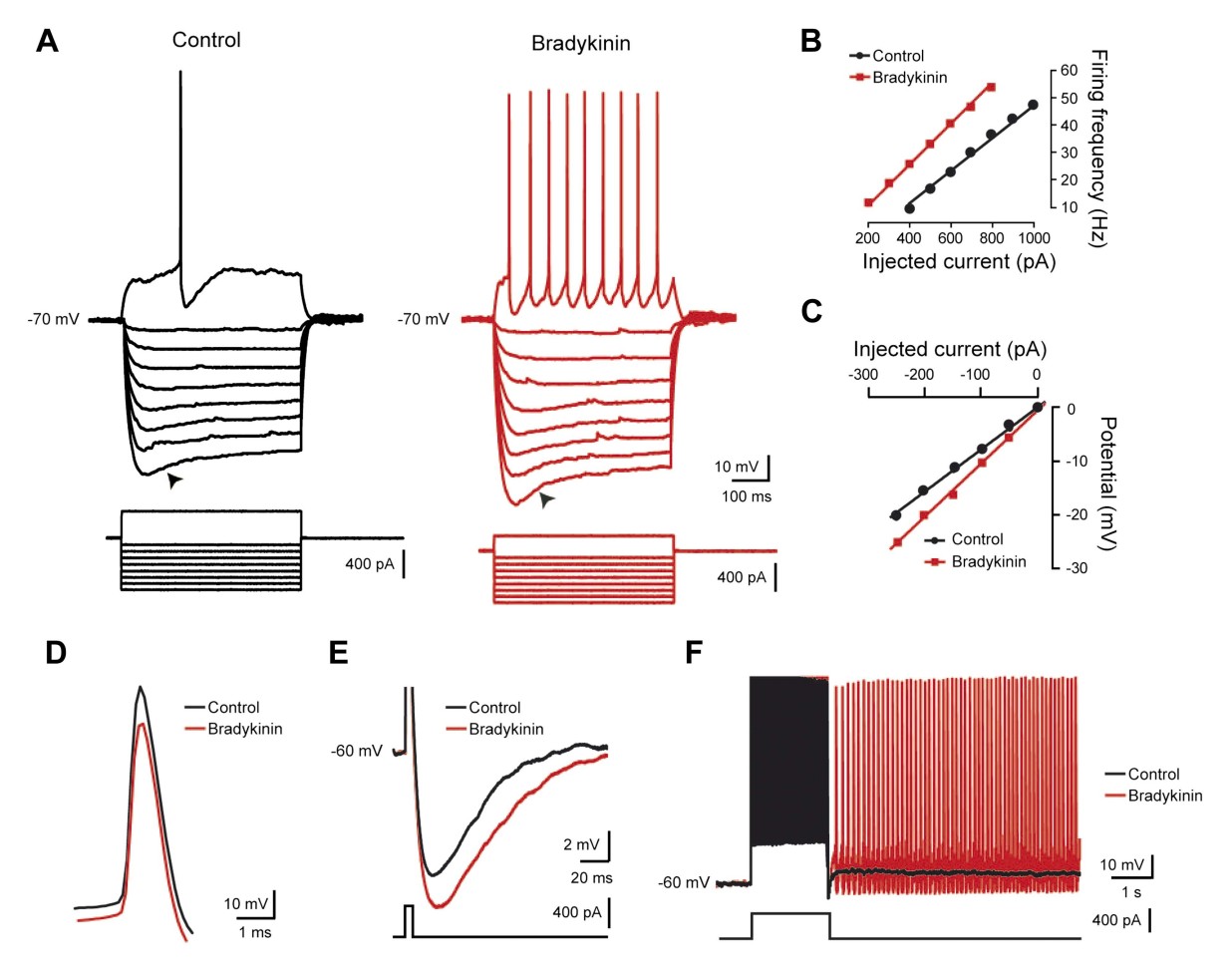

**Figure 2**. Bradykinin enhances repetitive firing and promotes self-sustained spiking. (**A–C**) Typical responses of a motoneuron to incrementing 1-s current injections (**A**) with its respective frequency–current (**B**) and voltage–current (**C**) relationships before (black traces) and after (red traces) bath applying bradykinin (Bk, 8 μM). Large hyperpolarizing currents revealed the presence of an inward rectification causing a depolarizing 'sag' (arrowheads). Initial potential was held at −70 mV. Note that a negative current injection was employed to counter the Bk-induced depolarization so that Bk measurements were also made at −70 mV. (**D**) Action potentials aligned in time on their peaks before and during Bk. (**E**) Afterhyperpolarization following a single action potential (truncated). Initial holding potential, −60 mV. (**F**) Superimposed voltage traces recorded in response to a 2-s depolarizing pulse from a holding potential of −60 mV before (black trace) and after (red trace) bath applying Bk (8 μM). Bottom traces in **A**, **E**, and **F** are the injected current protocol.

on lumbar motoneurons were secondary to the activation of NK1 receptors by substance P released from primary afferent C-fibers (*Dunn and Rang, 1990*). To test this hypothesis, we added Bk in the presence of spantide (2–5 μM), a selective NK1 receptor antagonist ($IC_{50}$ = 1.5 μM; *Merali et al., 1988*), that has been shown to depress the responses of neonatal rat motoneurons to substance P (*Yanagisawa and Otsuka, 1990*). Spantide did not attenuate Bk-induced responses (response in spantide 92.1 ± 18% of control response; p = 0.62, Wilcoxon paired test, n = 4 cells; see *Figure 3—figure supplement 1*). Note that the long-lasting reflex induced by Bk from an in vitro hemicord preparation was not occluded by spantide (2 μM) (number of events: 580 ± 58 for control vs 1859 ± 549 during Bk; p = 0.03, Wilcoxon paired test, n = 6; see *Figure 1—figure supplement 1*). Together, these results argue for a likely direct effect of Bk on lumbar motoneurons.

## Pharmacology of the Bk response

Bk exerts its biological effects through the activation of two receptors, called $B_1$ and $B_2$ (*Hall, 1992*). To determine which receptor subtype(s) mediate Bk's effects on motoneurons, we first examined the ability of selective agonists to reproduce the depolarization. Under TTX, the selective $B_1$ agonist,

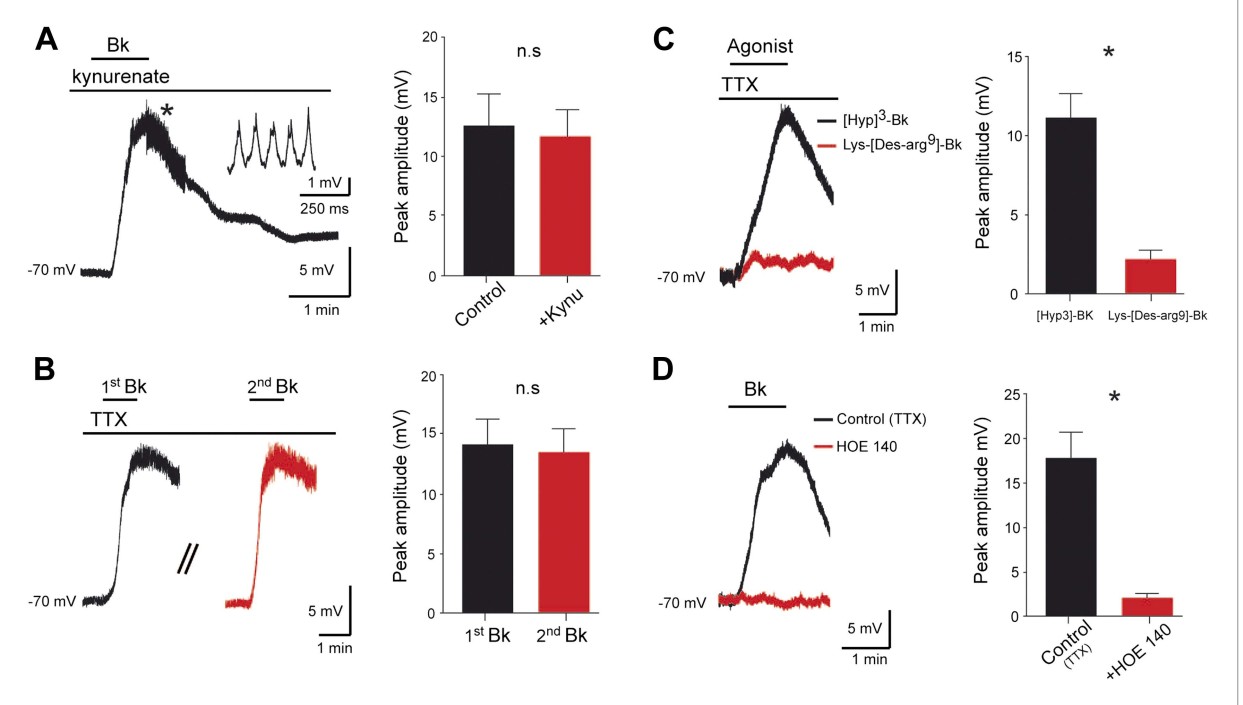

**Figure 3**. Bradykinin depolarizes lumbar motoneurons by a direct postsynaptic action of B2 receptors. (**A**) Voltage trace in response to bradykinin (Bk) collected under kynurenate (1.5 mM). The asterisk indicates the point shown by the trace at right with higher temporal resolution, where spikelets occurred. (**B**) Sequential depolarizations recorded under TTX (1 μM) induced by two successive applications of Bk (Bk, 8 μM) with intervals of 20 min (**C**) Voltage traces collected under TTX during application of either Lys-[Des-Arg9]-Bk, a selective B1 receptor agonist (red trace), or [Hyp3]-Bk, a selective B2 receptor agonist (black trace). (**D**) Voltage traces collected under TTX in response to bradykinin (Bk), before (black) and after (red) pretreatment with the selective B2 receptor antagonist HOE-140. At the right of each panel, graphs show the mean peak amplitude of membrane depolarizations induced by Bk or by one of its agonists. Drug application periods are indicated by lines above the records. Error bars indicate SEM. ns, not significant, *p < 0.05. (**A**, **B**, **D**: Wilcoxon paired test; **C**: Mann–Whitney test).

The following figure supplement is available for figure 3:

**Figure supplement 1**. At the left, superimposed voltage traces recorded from a L5 motoneuron in response to bradykinin (Bk) collected under TTX (0.5 μM) before and after the application of spantide (2 μM).

Lys-[Des-Arg$^9$]-Bk (Ki = 1.7 nM; *Marceau et al., 1998*), at concentrations as high as 3 μM, failed to induce a significant depolarization (2.2 ± 0.6 mV, n = 10 cells; *Figure 3C*). In contrast, the selective B$_2$ receptor agonist [Hyp]$^3$-Bk (Ki = 314 pM; *Windischhofer and Leis, 1997*), used at 2 μM evoked a membrane depolarization similar to responses evoked by Bk (11.2 ± 1.5 mV n = 10 cells for [Hyp]$^3$-Bk vs 12.9 ± 0.6 mV, n = 154 for Bk; p = 0.48, Mann–Whitney test; *Figure 3C*). Consistent with an involvement of B$_2$ receptors, the response to Bk was almost abolished by 2 μM HOE 140 (11.7 ± 2.7% of the control depolarization; p = 0.03, Wilcoxon paired test, n = 6 cells; *Figure 3D*), a potent and selective B$_2$ receptor antagonist (IC$_{50}$ = 420 pM; *Eggerickx et al., 1992*). Together, these results strongly suggest that Bk depolarizes motoneurons mainly via activation of B$_2$ receptors.

## Ionic basis of Bk actions

To determine the ionic mechanisms involved, motoneurons were held by voltage-clamp at −70 mV, a potential close to the normal resting membrane potential (*Tazerart et al., 2007*). To investigate the current in relative isolation, TTX (0.5 μM) was added in the aCSF. Under these conditions, Bk reversibly generated an inward current whose time course was similar to that seen in current-clamp recordings (*Figure 4A*). Part of the Bk response seems thus independent of voltage-gated channels. The inward current amplitude was steeply dependent on Bk concentration, with an EC$_{50}$ of 680 nM and maximal amplitude (322 ± 26 pA, n = 14) at 4–8 μM (*Figure 4B*). The peak current density was 1.5 ± 0.15 pA/pF

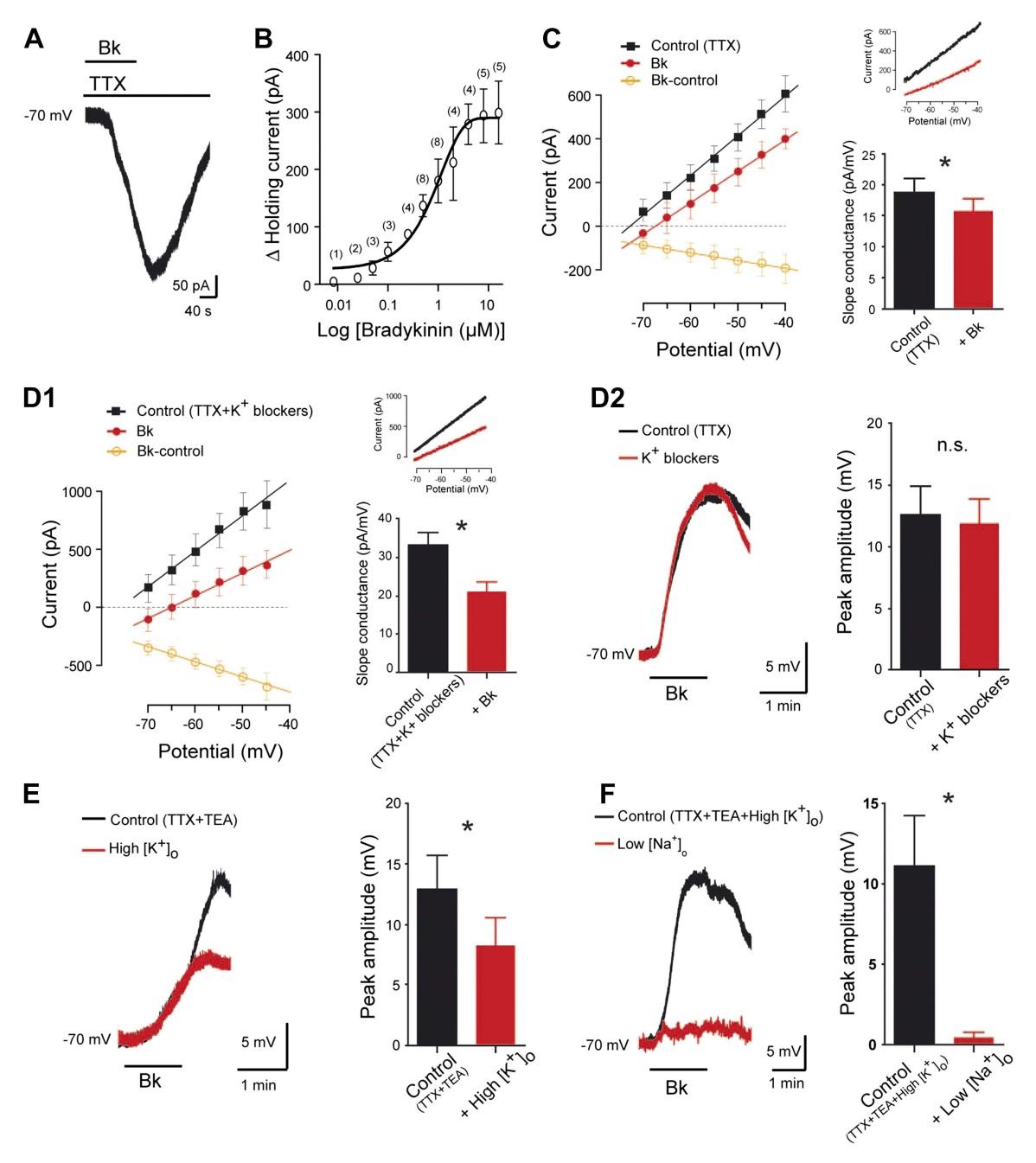

**Figure 4**. Bradykinin inhibits a leak K⁺ current and activates a Na⁺-dependent nonselective cationic current. (**A**) Representative inward current induced by bradykinin (Bk, 8 µM) in lumbar motoneuron. Voltage clamp, Holding potential, −70 mV. (**B**) Dose-response curve of Bk-induced changes in peak holding current. Numbers in the parenthesis are number of cells recorded. (**C**) At the left, I–V relationships reconstructed from voltage ramp data under TTX (0.5 µM) before (black trace) and at the peak of the response to bath-applied Bk (red trace). The orange trace with open circles illustrates the I–V relation of the difference current representing the Bk-sensitive current. At the right, current traces from a representative cell and histogram plotting changes in the slope conductance induced by Bk. (**D1**) At the left, I–V relationships reconstructed from voltage ramp data in a medium containing TTX (0.5 µM), TEA (10 mM), 4-aminopyridine (2 mM), the HCN-channel blocker ZD7288 (20 µM), and apamin (100 nM) before (black trace) and after (red trace) a bath application of Bk (8 µM). The orange trace with open circles illustrates the mean I–V relation of the difference current representing the Bk-sensitive current. At the right, current traces from a representative cell and histogram plotting changes in the slope conductance induced by Bk. (**D2**) Superimposed voltage traces under TTX (0.5 µM) and TEA (10 mM) in response to Bk before (black) and after (red) the superfusion of a medium containing TEA (10 mM), 4-aminopyridine (2 mM), the HCN-channel blocker ZD7288 (20 µM) and apamin (100 nM). (**E**) Superimposed voltage traces under TTX (0.5 µM) and TEA (10 mM) in response to Bk before (black) and after (red) the superfusion of a medium with high $[K^+]_o$ (9 mM). (**F**) Superimposed voltage traces under TTX
*Figure 4. continued on next page*

*Figure 4. Continued*

(0.5 μM), TEA (10 mM), and high $[K^+]_o$ (9 mM) in response to Bk before (black) and after (red) the superfusion of a medium with low $[Na^+]_o$. At the right of each panel, histogram plotting the peak amplitude of membrane depolarizations induced by Bk before (black) and after (red) the superfusion of the medium. Drug application periods are indicated by horizontal lines. Error bars indicate SEM, ns, not significant, *p < 0.05 (Wilcoxon paired test).

(n = 14). In the following experiments, the Bk-sensitive current was investigated by means of slow voltage ramp to minimize inactivating voltage-gated currents. To ensure a maximum response and to avoid potential fluctuation of the effective peptide concentration, Bk was applied at 8 μM. A significant Bk-induced decrease in the slope conductance was observed in *I/V* curves (18.7 ± 2.4 pA/mV before Bk vs 15.6 ± 2.2 pA/mV during Bk; p = 0.016, Wilcoxon paired test, n = 7 cells; *Figure 4C*), an effect often interpreted as a consequence of the inhibition of a $K^+$ conductance. Consistent with this interpretation, the linear *I/V* relationship of the Bk-sensitive-current (obtained by subtracting the control from the Bk currents: *Figure 4C*, *orange line with open circles*) reversed at an extrapolated value close to the predicted equilibrium potential for $K^+$ ($E_k$: −100 mV). The combination of traditional voltage-gated $K^+$ channel blockers [TEA (10 mM), 4-aminopyridine (2 mM), the HCN-channel blocker ZD7288 (20 μM)] and the $Ca^{2+}$-activated $K^+$ channel blocker apamin (100 nM), did not change either the Bk-induced decrease in slope conductance (33.2 ± 3.2 pA/mV before Bk vs 21.1 ± 2.3 pA/mV during Bk; p = 0.02, Wilcoxon paired test, n = 15 cells; *Figure 4D1*) or the peak depolarization magnitude measured in current clamp from −70 mV (94.5 ± 15.8% of the control; p = 0.54, Wilcoxon paired test, n = 7 cells; *Figure 4D2*). The contribution of $K^+$ ions was further examined by raising the extracellular $K^+$ ($[K^+]_o$) to 9 mM, a manipulation that almost eliminates the driving force for $K^+$ at resting membrane potential by setting $E_k$ at ∼ −70 mV according the Nernst equation. Under high $[K^+]_o$, Bk still induced a depolarization from −70 mV without a measurable change in input resistance (45.3 ± 13.9 MΩ before Bk vs 46.5 ± 13.2 MΩ during Bk; p = 0.81, Wilcoxon paired test, n = 7 cells). However, the depolarization was significantly smaller than that seen in normal aCSF (64.2 ± 17.7% of the control value measured in normal aCSF; p = 0.047, Wilcoxon paired test, n = 7 cells; *Figure 4E*). These data suggest that a second ionic component contributes to the Bk depolarization in addition to a reduction in $K^+$ conductances. The residual Bk-induced depolarization observed in high $[K^+]_o$ appears to be sodium-mediated because it disappeared after reducing $[Na^+]_o$ from 152 to 26 mM with equimolar choline-chloride (4.1 ± 2.8% of the control value measured in high $[K^+]_o$; p = 0.016, Wilcoxon paired test, n = 7 cells; *Figure 4F*). In sum, our data indicate that there are at least two ionic components in the response to Bk: a $K^+$ component that produces an input resistance increase and likely involves a reduction of a resting $K^+$ conductance, and a $Na^+$ component that produces depolarization without measurable change in input resistance.

## Reduction of a $Ba^{2+}$-sensitive $K^+$ leak conductance accounts for part of the Bk-induced depolarization

The foregoing results suggest that reduction of a resting $K^+$ conductance associated with an increase of the input resistance might help Bk to generate the slow depolarization. To test the contribution of a resting $K^+$ conductance, we used quinidine (200–400 μM), a broad-spectrum blocker of neuronal background $K^+$ channels ($IC_{50} \approx 100$ μM; *Lesage and Lazdunski, 2000*). Quinidine induced alone a large depolarization (+6.8 ± 3.9 mV, n = 6 cells, not shown) and reduced the slope conductance itself (22 ± 3.8 pA/mV before quinidine vs 14.2 ± 3.4 pA/mV during quinidine; p < 0.01, One-way ANOVA test, n = 6 cells; *Figure 5A1*). Further, quinidine prevented the slope conductance decrease induced by Bk (14.2 ± 3.4 pA/mV for control vs 13 ± 3 pA/mV during Bk; p > 0.05, One-way ANOVA test, n = 6; *Figure 5A1*) and partially occluded the Bk-induced membrane depolarization (57.7 ± 21.5% of control; p = 0.03, Wilcoxon paired test, n = 6; *Figure 5A2*). These results show that the blockade of a resting $K^+$ conductance can both mimic and partially occlude the effects of Bk.

We further characterized the nature of the resting $K^+$ conductance modulated by Bk. Because a fraction of background $K^+$ channels is downregulated by extracellular acidosis (*Talley et al., 2000*; *Patel and Honore, 2001*), motoneurons were exposed to Bk before and after lowering the pH from 7.4 to 6.4. As expected, acidification induced a membrane depolarization (+4.3 ± 0.6 mV, n = 5 cells, not shown) but did not significantly decrease the slope conductance itself (26.4 ± 6.9 pA/mV at pH 7.4 vs 21.2 ± 5.2 pA/mV at pH 6.4; p > 0.05, One-Way ANOVA test, n = 5 cells; *Figure 5B1*). Furthermore,

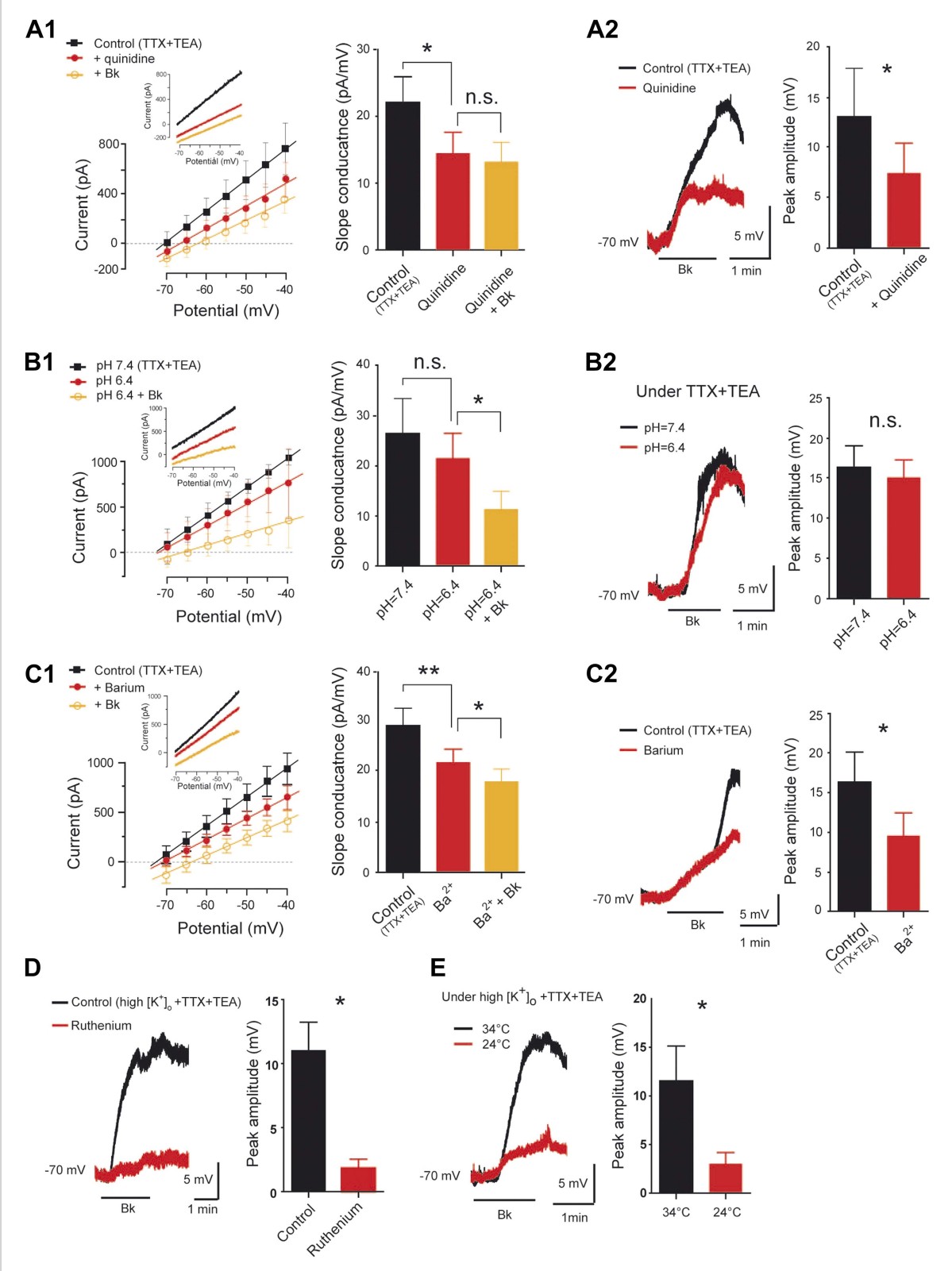

**Figure 5.** Pharmacological profile of the bradykinin-induced current. (**A1**–**C1**) At the left, superimposed I–V relationships reconstructed from voltage ramp data (representative data in inserts) before (black trace) and after (red trace with filled circles) quinidine (200 µM, **A1**), extracellular acidosis (**B1**), or barium (300 µM, **C1**). In each panel the orange trace with open circles illustrates the mean I–V relation obtained during bradykinin application (Bk, 8 µM) under

*Figure 5. Continued*

quinidine (**A1**), extracellular acidosis (**B1**), or barium (**C1**). At the right of each panel, histogram plotting changes in the slope conductance. (**A2–C2**) At the left, superimposed voltage traces in response to Bk before (black) and after (red) quinidine (200 µM, **A2**), extracellular acidosis (**B2**), or barium (300 µM, **C2**). At the right of each panel, histogram plotting the peak amplitude of membrane depolarizations induced by Bk. (**D–E**) At the left, superimposed voltage traces under high $[K^+]_o$ in response to Bk before (black) and after (red) applying ruthenium red (200 µM, **D**) or lowering temperature from 34 to 24°C (**E**). At the right of each panel, histogram plotting the peak amplitude of membrane depolarizations induced by Bk. All recordings were performed under TTX (1 µM) and TEA (10 mM). Error bars indicate SEM, *p < 0.05, **p < 0.01 (**A1**, **B1**, **C1**: One-Way ANOVA; **A2**, **B2**, **C2**, **D**, **E**: Wilcoxon paired test).

the response to Bk was not affected by acidification to pH 6.4, that is, the slope conductance decrease (21.2 ± 5.2 pA/mV for control vs 11.2 ± 3.6 pA/mV during Bk; p < 0.05, One-Way ANOVA test, n = 5 cells; *Figure 5B1*) and the membrane depolarization from −70 mV were not affected (92.6 ± 16.0% of values at pH 7.4; p = 0.69, Wilcoxon paired test, n = 6 cells; *Figure 5B2*). We next tested $Ba^{2+}$, which is known to block a subset of resting $K^+$ currents. The Bath application of $Ba^{2+}$ (300 µM) caused a significant depolarization (+6.6 ± 0.8 mV, n = 6 cells, not shown) and also by itself decreased the slope conductance (29.3 ± 3.2 pA/mV for control vs 22 ± 2.6 pA/mV during barium; p < 0.01, One-Way ANOVA test, n = 6 cells; *Figure 5C1*). $Ba^{2+}$ reduced but did not prevent the slope conductance decrease induced by Bk (22 ± 2.6 pA/mV for control vs 18.1 ± 2.4 pA/mV during Bk; p < 0.05, One-Way ANOVA test, n = 6; *Figure 5C1*), and it occluded the Bk-induced membrane depolarization from −70 mV (58.6 ± 17.7% of control; p = 0.03, Wilcoxon paired test, n = 6 cells; *Figure 5C2*). In sum, it appears that a $Ba^{2+}$-sensitive, pH-insensitive resting $K^+$ conductance accounts for a substantial fraction of the $K^+$ component modulated by Bk.

## A nonselective cationic conductance accounts for part of the Bk-induced depolarization

In addition to Bk's inhibition of a resting $K^+$ current, we also characterized the activation of the conductance carried by $Na^+$. We recently characterized a thermosensitive nonselective cationic current ($I_{CaN}$) in lumbar motoneurons of neonatal rats (*Bouhadfane et al., 2013*). To determine the contribution of $I_{CaN}$ to the Bk-evoked depolarization, we examined the effect of ruthenium red, routinely used to block channels that engender $I_{CaN}$ (*Wu et al., 2010*). In high $[K^+]_o$, the Bk-induced depolarization from −70 mV (where the effect of the resting $K^+$ current is minimized) was nearly abolished by 10 µM ruthenium red (16.6 ± 6.5% of control response; p = 0.03, Wilcoxon paired test, n = 6 cells; *Figure 5D*). This apparent $I_{CaN}$ was also thermosensitive because lowering the temperature from 34 to 24°C induced a significant decrease in the Bk-induced depolarization at −70 mV in high $[K^+]_o$ (26.2 ± 9.8% of control values; p = 0.03, Wilcoxon paired test; *Figure 5E*). In sum, it appears that a current with properties of a thermosensitive $I_{CaN}$ might account for part of the Bk-induced depolarization.

## Bk-induced depolarization is related to a rise in intracellular $Ca^{2+}$ in dendrites

Increases in the concentration of free cytosolic $Ca^{2+}$ ($[Ca^{2+}]_i$) are fundamental in recruiting $I_{CaN}$ from neonatal rat lumbar motoneurons (*Bouhadfane et al., 2013*). We therefore examined whether an increase in intracellular calcium ($[Ca^{2+}]_i$) was required for the Bk-induced depolarization. To demonstrate a Bk-induced rise of $[Ca^{2+}]_i$ in motoneurons, we first loaded the ventral horn with the calcium indicator Oregon Green-BAPTA 1 AM (OGB1-AM) along with the glial cell marker Sulforhodamine-101 (SR-101) using the extracellular bolus loading technique (see 'Materials and methods'). Since the astroglial cells were labeled by both dyes, they appeared as yellow cells, while neurons were labeled by green OGB1-AM only. This allowed the clear separation of neurons from astroglial cells. After a brief equilibration period, we imaged calcium changes from somata of lumbar motoneurons using a three-dimensional random-access two-photon microscope (*Figure 6A*). Bk induced a rise in $[Ca^{2+}]_i$ (13.4 ± 0.7% ΔF/F) in 65% of motoneuron somata (n = 177 out of 272 cells; n = 5 slices), with very slow decay times (*Figure 6B,C*); there was no change in $[Ca^{2+}]_i$ in the remaining motoneurons. In the next step, we explored the relationship between spatio-temporal changes in $[Ca^{2+}]_i$ in motoneuron somata and dendrites along with the Bk-induced membrane

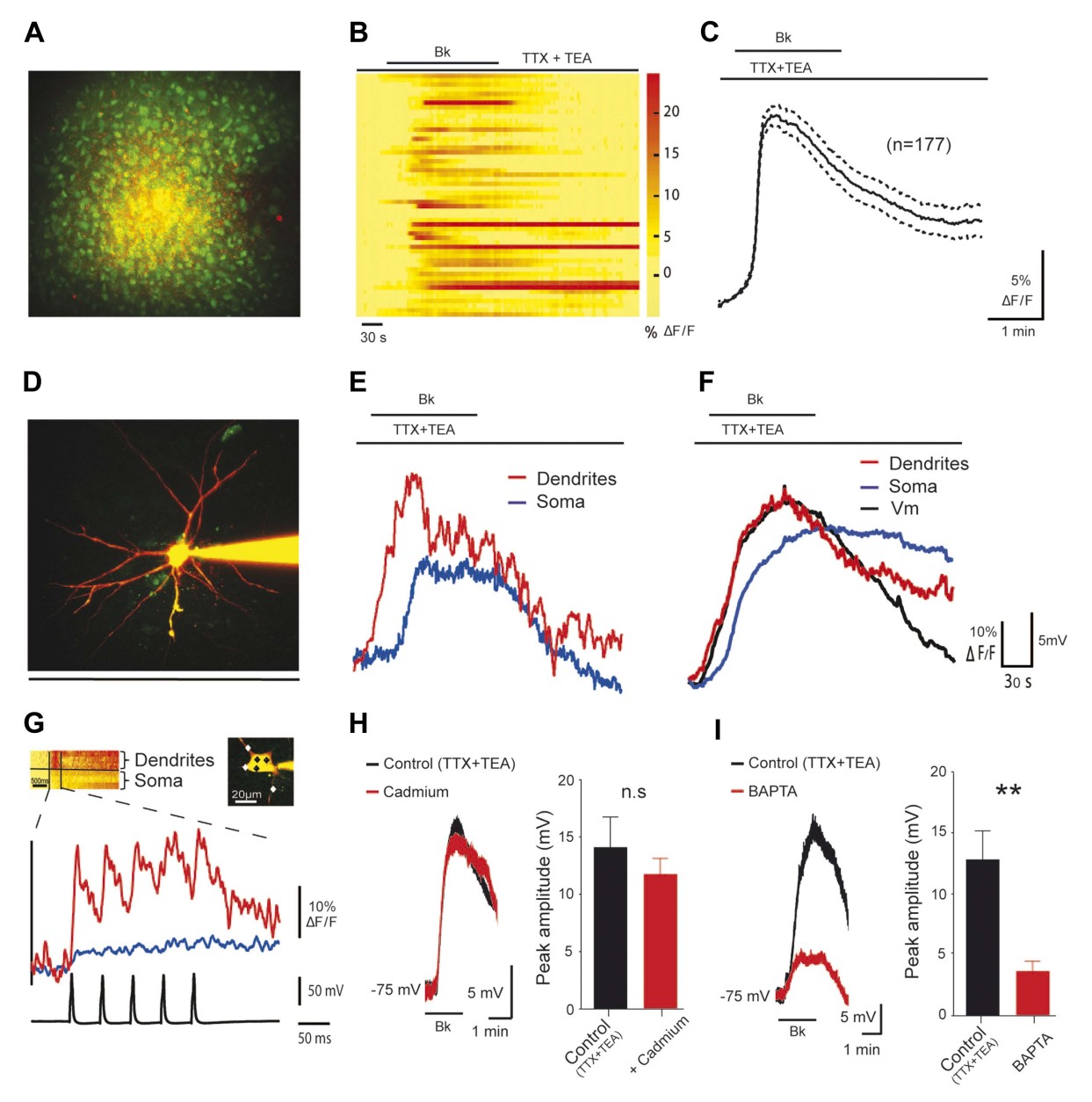

**Figure 6**. The Bk-induced depolarization of lumbar motoneurons is associated with $[Ca^{2+}]_i$ rise in dendrites. (**A**) z-projection of two-photon image stack showing cells loaded by bolus injection of OGB1-AM (1 mM) and SR-101 (300 µM). (**B**) Bradykinin (Bk, 8 µM) increases calcium levels in the motoneuron somata. Each line is a different motoneuron, aligned to the time of Bk application, see color code (right) for quantification of increase in calcium fluorescence (**C**) Time course of the mean (±SEM) population calcium response to Bk application; n = 177 neurons. (**D**) z-projection of a two-photon image stack showing a representative motoneuron loaded with OGB1 (120 µM) and Alexa 594 (20 µM). (**E**) Simultaneously recorded dendritic (red trace) and somatic (blue trace) calcium transients of the motoneuron shown on (**D**). (**F**) Average dendritic (red trace) and somatic (blue trace) calcium responses along with the simultaneously recorded average voltage trace (black trace) acquired from the patched cells. n = 5 neurons. (**G**) Average somatic (blue trace) and dendritic (red trace) calcium traces of 5 sweeps in response to action potentials evoked by somatic current injection. Inset, points of interest on soma and dendrites (black and white diamonds, respectively). (**H–I**) At the left, superimposed voltage traces under TTX (0.5 µM) and TEA (10 mM) in response to Bk (Bk, 8 µM) before (black) and after (red) the superfusion of (**H**) cadmium (200 µM, **H**) or intracellular diffusion of BAPTA (10 mM, **I**). At the right of each panel, histogram plotting the peak amplitude of membrane depolarizations induced by Bk before (black) and after (red) the drug application. Error bars indicate SEM, **p < 0.01 (Wilcoxon paired test).

depolarization. Here, the patch pipette was filled with the cell-impermeable calcium indicator OGB1 (120 µM) and a morphometric dye, Alexa 594 (20 µM) to allow simultaneous two-photon imaging and anatomical tracing of the cellular processes (*Figure 6D*). After Bk application, the proximal dendrites up to 50 µm away from the soma showed a clear rise in $[Ca^{2+}]_i$ (49.8 ± 8.7% ΔF/F; n = 5 cells; *Figure 6E,F*; *red trace*), which was followed with a delay by the somatic response (44.5e ± 8.9% ΔF/F; n = 5 cells; *Figure 6E,F*; *blue trace*). All of the above results were obtained with blocked voltage-gated sodium and potassium channels (0.5 µM TTX and 10 mM TEA, respectively), showing the Bk-induced response alone, without spiking activity of the motoneurons. To obtain a physiologically relevant comparison to the Bk-induced responses, we imaged $[Ca^{2+}]_i$ changes induced by action potentials evoked by depolarizing pulses in the absence of TTX and TEA (*Figure 6G*). Our results show that a single action potential induced a $[Ca^{2+}]_i$ rise significantly higher in the proximal dendrites than in the soma ($Ca_{soma}$ = 6.1 ± 1.5% ΔF/F; $Ca_{dendrite}$ = 16.9 ± 3.6% ΔF/F; n = 5 cells; p = 0.02, Student's paired *t*-test). The Bk-induced rise in $[Ca^{2+}]_i$ was significantly higher than that evoked by a single action potential in both the soma and the dendrites (p = 0.003 and p = 0.008, Student's unpaired *t*-test). In conclusion, the data obtained with calcium imaging show that Bk induces a significant increase in $[Ca^{2+}]_i$.

## Increases in $[Ca^{2+}]_i$ are caused by $Ca^{2+}$ released from intracellular stores

We dissected the sources of the Bk-induced increase in $[Ca^{2+}]_i$. Calcium can enter the cytoplasm via several routes including the NMDA-receptor channels, voltage-gated calcium channels and release from internal calcium stores. Blockade of NMDA-receptor channels with kynurenate (1.5 mM) or CNQX (10 µM) and AP-5 (20 µM) did not prevent the depolarization induced by Bk (*Figure 3B*). Similarly, the effects of Bk on motoneurons were not significantly reduced by cadmium blockade of voltage-gated calcium channels (100 µM; 83 ± 9.7%, of control depolarization; p = 0.46, Wilcoxon paired test, n = 8 cells; *Figure 6H*). The dependence of the depolarization on $[Ca^{2+}]_i$ was then studied by recording motoneurons after intracellular perfusion of the $Ca^{2+}$ chelating agent BAPTA (10 mM) from the pipette solution. In these experiments, we applied Bk as soon as possible after establishing whole-cell access, so that we could record a control response before full diffusion of BAPTA into the cell. The first application caused a robust depolarization similar to that in the control cells (12.7 ± 2.4 mV, n = 8 cells vs 12.9 ± 0.6 mV for control, n = 154 cells, p = 0.94, Mann–Whitney test). 20 minutes after establishing the whole cell recording, the depolarization induced by Bk declined to 28.8 ± 6.4% of its original value (p = 0.008, Wilcoxon paired test, n = 8 cells; *Figure 6I*). Combined with our calcium imaging results, these experiments suggest that the Bk-induced increase in $[Ca^{2+}]_i$ observed in our experiments was likely caused by $Ca^{2+}$ released from intracellular stores rather than by $Ca^{2+}$ influx through $Ca^{2+}$-permeable membrane channels.

## Bk effects are mediated by the G-protein/PLC/InsP3 pathway

To test whether a G-protein-mediated pathway is involved, we diffused the nonhydrolyzable GDP analog GDPβS (2 mM), a broad spectrum G-protein inhibitor, from the pipette. Immediately after breakthrough, before the passive loading of GDPβS into the cell, the first application of Bk induced the expected transient slow depolarization (15.7 ± 2.4 mV, n = 6 cells). After diffusion of GDPβS into the cell, the subsequent application of Bk failed to induce a depolarization (9.7 ± 5.7% of control, p = 0.03, Wilcoxon paired test, n = 6 cells; *Figure 7A*). Bk receptors are generally described as signaling through $G_q$ α-subunit (*Gutowski et al., 1991*; *LaMorte et al., 1993*), but they can also interact with other G proteins such as $G_s$ (*Liebmann et al., 1996*) and $G_i$ α-subunits (*Ewald et al., 1989*; *Yanaga et al., 1991*). We showed that Bk signaling is independent of $G_i$ or $G_s$ protein by showing that after 7 hr of preincubation with either the $G_i$ inhibitor pertussis toxin (PTX, 2 µg.ml$^{-1}$) or the $G_s$ inhibitor cholera toxin (CTX, 2 µg.ml$^{-1}$) the magnitude of Bk responses was not disturbed (CTX: 14.7 ± 2.7 mV, n = 5 cells, p = 0.6; PTX: 16.7 ± 1.8 mV, n = 4 cells p = 0.3; control: 12.9 ± 0.6 mV, n = 54 cells, Mann–Whitney test; data not shown). These data suggest the contribution of $G_q$ to the transduction process.

Classically, $G_q$ proteins are coupled to a phospholipase C (PLC) signaling pathway (*Selbie and Hill, 1998*). To test whether Bk acts by this pathway, we used the PLC inhibitor U73122 (IC$_{50}$ = 5 µM; *Bleasdale et al., 1990*). U73122 (10 µM) greatly reduced Bk-induced depolarization (24.6 ± 4% of control; p = 0.03, Wilcoxon paired test, n = 6 cells; *Figure 7B*). The PLC pathway bifurcates to generate inositol 1,4,5-trisphosphate (InsP3) and calcium release from InsP3-sensitive intracellular

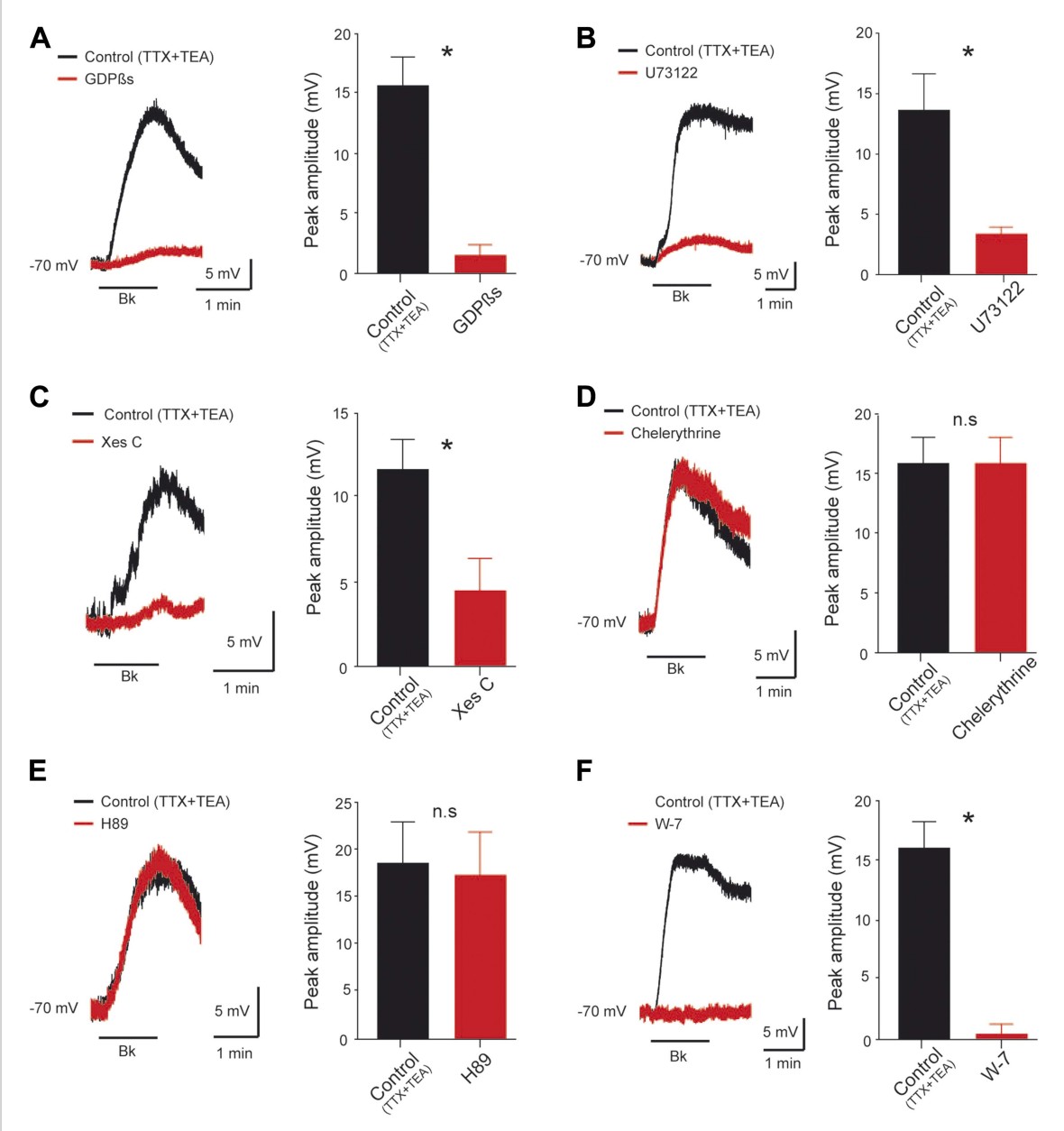

**Figure 7**. Signal transduction mechanism underlying the Bk effects. (**A–F**) At the left, superimposed voltage traces under TTX (0.5 μM) and TEA (10 mM) in response to bradykinin (Bk, 8 μM) before (black) and after (red) GDPßS (2 mM, **A**), U73122 (10 μM, **B**), Xestospongin C (2.5 mM, **C**), Chelerythrine (10 μM, **D**), H89 (10 μM, **E**) or W-7 (100 μM, **F**). At the right of each panel, histogram plotting the peak amplitude of membrane depolarizations induced by Bk before (black) and after (red) the drug application. Error bars indicate SEM, *p < 0.05 (Wilcoxon paired test).

stores on the one hand, and production of diacylglycerol (DAG) which activates protein kinase C (PKC) on the other. We tested the role of InsP3 by adding xestospongin C, an alkaloid InsP3 receptor antagonist (IC$_{50}$ = 0.4 μM; *Gafni et al., 1997*). Intrapipette loading with xestospongin (2.5 mM) induced a significant reduction of the Bk-induced depolarization (38.8 ± 17.4% of control, p = 0.015, Wilcoxon paired test, n = 8 cells, *Figure 7C*). In contrast, bath application of chelerythrine (10 μM) or intracellular diffusion of PKC 19–36 (10 μM), two potent inhibitors of PKC (IC$_{50}$ = 0.66 μM and 0.3 μM respectively; *Herbert et al., 1990*; *Smith et al., 1990*), did not reduce the Bk response (pooled data: 10.5 ± 17.8% of control, p = 1, Wilcoxon paired test, n = 5 cells 3 with chelerythrine and 2 with PKC

19–36, *Figure 7D*) indicating that the DAG pathway is not involved. Similar results were collected with either 10 µM bath application of H89 or 10 µM intracellular administration of PKA 6–22 (95.9 ± 12.9% of control, p = 0.31, Wilcoxon paired test, n = 6 cells 3 with H89 and 3 with PKA 6–22, *Figure 7E*), two potent blockers of PKA ($IC_{50}$ = 0.14 µM and 2 nM respectively; (*Glass et al., 1989*; *Davies et al., 2000*).

Multiple $Ca^{2+}$-sensing proteins detect the elevation of free cytosolic $Ca^{2+}$ and become downstream elements in intracellular calcium signaling pathways. Among them, calmodulin (CaM), a small cytoplasmic $Ca^{2+}$-sensing protein, regulates many ion channels in a $Ca^{2+}$-dependent manner. Supporting a role for CaM in the Bk response, application of W-7 (100 µM), a membrane permeable CaM inhibitor ($IC_{50} \approx 30$ µM; *Hidaka et al., 1981*), abolished the Bk-induced depolarization (3.1 ± 5%; p = 0.02, Wilcoxon paired test, n = 6 cells; *Figure 7F*). This result, and the requirement for intracellular calcium build-up, suggested that the Bk-induced depolarization may be mediated by $Ca^{2+}$/CaM-dependent protein kinase II (CaMKII). To test this hypothesis, we inhibited CaMKII by preincubating the neurons (≥90 min) with KN-93 (10 µM), a potent CaMKII inhibitor ($IC_{50}$ = 0.4 µM; *Sumi et al., 1991*). However, the Bk-evoked motoneuron depolarization was not significantly affected by KN-93 (KN-93: 10.9 ± 2.4 mV, n = 6 cells; Control: 12.9 ± 0.6 mV, n = 154; p = 0.52, Mann–Whitney test, data not shown). In sum, Bk appears to inhibit a leak $K^+$ conductance and activate $I_{CaN}$ via a transduction system activating PLC-/InsP3-mediated release of intracellular $Ca^{2+}$ and CaM.

## Discussion

Here we provide the first evidence supporting the possibility of a direct excitation of rat spinal motoneurons by Bk. Acting mainly through $B_2$ receptors, Bk acts by at least two ionic mechanisms involving a reduction of a resting $K^+$ conductance and activation of a nonselective cationic conductance mediated primarily by $Na^+$. This process is initiated by a $[Ca^{2+}]_i$ rise in dendrites triggered by a G-protein-dependent PLC-InsP3 signaling pathway.

### Action of Bk on motoneurons

The excitatory effects of Bk on motoneurons are reminiscent of its action on in vitro spinal cord preparations from neonatal rats where it slowly depolarizes ventral roots (*Dray et al., 1988*; *Dunn and Rang, 1990*; *Dray et al., 1992*; *Rueff and Dray, 1993*); see also (*Figure 1B*). The actions of Bk in generating motor outputs have long been viewed as secondary to release of substance P from capsaicin-sensitive sensory neurons (*Bynke et al., 1983*; *Geppetti, 1993*; *Jeftinija, 1994*). This indirect mechanism for exciting motoneurons is not used since the substance P receptor antagonist spantide did not antagonize responses to Bk in our study. Our results suggest a direct postsynaptic action of Bk on motoneurons that is still seen when sensory inputs are blocked by TTX or cadmium. Further supporting this, Bk receptors have been mapped to the membranes of motoneurons (*Lopes et al., 1995*), Bk induces direct elevation of free calcium in motoneurons, and the depolarization in motoneurons is greatly reduced by calcium chelators infused into the motoneurons themselves. On the other hand, microglial cells express functional Bk receptors (*Noda et al., 2003*; *Ifuku et al., 2007*) and Bk can act as a neuron-glia signaling molecule (*Parpura et al., 1994*; *Heblich et al., 2001*). Therefore, we cannot be rule out the possibility that bath-applied Bk activates glial cells resulting in the release of an unidentified chemical substance which may mediate some of Bk's effects on motoneurons.

### Functional evidence of a role for $K_{2P}$ and TRP channels

Bk has long been shown to depolarize sensory neurons by inhibiting $K^+$ channels (*Weinreich, 1986*; *Brown and Higashida, 1988*; *Jones et al., 1995*; *Villarroel, 1996*; *Cruzblanca et al., 1998*; *Liu et al., 2010*). We have found that inhibition of $K^+$ channels seems also to occur in motoneurons, as seen by a decrease of the slope conductance which reverses near $E_k$ and is influenced by changes in $[K^+]_o$ and addition of the $K^+$ blocker $Ba^{2+}$. The Bk-sensitive $K^+$ current might be derived from a member(s) of the two-pore domain $K^+$ ($K_{2P}$) channels that are substrates for resting $K^+$ currents in neurons (*Enyedi and Czirjak, 2010*). This structurally distinct group of $K^+$ channels includes over 15 members classified into six functional subgroups: (1) TWIK-1 and -2; (2) TASK-1, -3 and -5; (3) TREK-1, and -2 and TRAAK; (4) TASK-2, TALK-1 and -2; (5) THIK-1; (6) TRESK. Functional distinction between $K_{2P}$ channel subtypes is still a delicate issue because of a lack of selective pharmacological reagents (*Lesage, 2003*; *Lotshaw, 2007*). However, several features help to separate them (*Patel and Honore, 2001*). In addition to quinidine sensitivity, we show that the response to Bk is reduced by a low concentration

of $Ba^{2+}$ but not by acidosis. The relatively weak pH sensitivity of the response is not consistent with the properties of TASK, TALK, and TRESK channels that are downregulated by acidosis (*Duprat et al., 1997*; *Sano et al., 2003*; *Kang and Kim, 2004*; *Dobler et al., 2007*; *Mathie et al., 2010*; *Callejo et al., 2013*). Likewise, the contribution of TREK/TRAAK channels is unlikely because they are weakly inhibited by or resistant to $Ba^{2+}$ (*Lesage et al., 2000*; *Patel and Honore, 2001*). Thus the most likely candidates for the $Ba^{2+}$-sensitive $K^+$ current are TWIK and THIK channels (*Chavez et al., 1999*; *Patel et al., 2000*; *Rajan et al., 2001*; *Lloyd et al., 2009*) though TWIK channels are much more sensitive to $Ba^{2+}$ ($IC_{50}$: 0.1 mM) than THIK ($IC_{50}$: ~1 mM) (*Patel and Honore, 2001*; *Rajan et al., 2001*). The contribution of these channels in generating the Bk effects remains to be established.

While it is clear that Bk inhibits a $Ba^{2+}$-sensitive $K^+$ current, it is also clear that it activates a $Ba^{2+}$-insensitive $Na^+$ conductance. This conductance shares features of $I_{CaN}$ (*Partridge and Swandulla, 1988*), as it appears to be: (1) voltage-independent, (2) calcium-activated, (3) blocked by the cationic channel blocker ruthenium red, and (4) eliminated by lowering extracellular $Na^+$. The fact that the $Na^+$-dependent conductance mediates a significant current without affecting the input resistance suggests that it may have a dendritic location and help regulate the flow of distal synaptic inputs to the soma. The initial rise of $[Ca^{2+}]_i$ at the dendritic level supports this hypothesis and is consistent with the location of $B_2$ receptors in the peripheral processes of motoneurons (*Lopes et al., 1995*). Channels capable of mediating a $Na^+$ response similar to that reported in this study are mammalian homologues of *Drosophilia* transient receptor potential (TRP) channels (*Wu et al., 2010*). Consistent with this, Bk has been shown to activate different classes of TRP channels (*Premkumar and Ahern, 2000*; *Delmas et al., 2002*; *Sugiura et al., 2002*; *Bandell et al., 2004*; *Zhang et al., 2008*) and to induce a $Na^+$-mediated $I_{CaN}$ in cultured spinal sensory neurons (*Burgess et al., 1989*; *Dray et al., 1992*; *McGehee et al., 1992*; *Seabrook et al., 1997*). The cationic current could be produced by gating of more than a single class of channels. Given the sensitivity of Bk-induced responses to temperature, thermo-TRP channels (including TRPV and TRPM) might be activated by Bk in motoneurons. Still, it should be stressed that additional studies will be necessary to precisely define the channel type(s) that mediate Bk responses.

Although Bk appears to act by at least two ionic mechanisms in motoneurons, activation of a single $B_2$ receptor class appears to occur: $B_2$ (but not $B_1$) receptors are expressed in motoneurons (*Lopes et al., 1995*), and $B_2$ agonists and antagonists mimic or block all the effects of Bk.

## Signaling cascades

Several groups have provided data in different systems suggesting that Bk depolarizes neurons by triggering a PLC signaling pathway giving rise to the InsP3-dependent release of $Ca^{2+}$ and DAG (*Yano et al., 1984*, *1985*; *Derian and Moskowitz, 1986*; *Francel and Dawson, 1986*; *Francel et al., 1987*; *Jackson et al., 1987*; *Thayer et al., 1988*; *Perney and Miller, 1989*; *Gutowski et al., 1991*; *Hall, 1992*; *Premkumar and Ahern, 2000*; *Ferreira et al., 2004*). Our pharmacological data support this pathway, as the Bk response is: (1) blocked by GDPβS; (2) blocked by the PLC antagonist U73122; (3) blocked by the InsP3 receptor antagonist xestospongin C. The DAG arm of the bifurcating PLC pathway appears not to be involved, as the Bk-induced depolarization was not inhibited by PKC antagonists.

Consistent with a PLC/InsP3 pathway, we found that a rise of intracellular $Ca^{2+}$ is essential for Bk responses. Since blockade of membrane $Ca^{2+}$ channels does not affect Bk-induced depolarizations, and $I_{CaN}$ is mainly mediated by $Na^+$, the $[Ca^{2+}]_i$ increase must arise from the release of $Ca^{2+}$ from intracellular stores. Most directly, we showed that the introduction of the $Ca^{2+}$ buffer BAPTA into the cytoplasm greatly reduced the Bk response, and blockade of the $Ca^{2+}$ sensor CaM fully abolished Bk responses. As summarized in *Figure 8*, these data suggest that the two ionic components are mediated by $B_2$ receptors acting via the single PLC/InsP3/calcium signaling pathway.

## Functional consequences

Bk not only depolarizes spinal motoneurons but also sensitizes them, so they are more activated by small inputs than under control conditions. The leftward shift in the *f–I* relation can be explained as an increase in responsiveness due to an increase in input resistance. The depolarizing effect of the $Na^+$ current would also be enhanced by the simultaneous increase in input resistance due to reduction of a resting $K^+$ conductance. This synergistic mechanism might promote the burst-evoked sADP, which helps to generate self-sustained spiking (*Bouhadfane et al., 2013*). Following traumatic injury, increases in the concentration of kinins in the spinal cord (*Xu et al., 1991*) might thus predispose

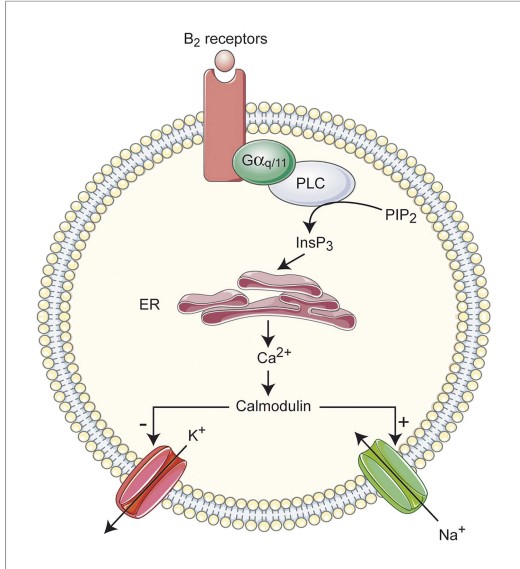

**Figure 8**. Overview of the signal transduction cascade for excitatory actions of Bk on lumbar motoneurons. InsP3, inositol 1,4,5-trisphosphate; PIP2, phosphatidyli-nositol-4,5-diphosphate; PLC, Phospholipase C; ER, endoplasmic reticulum.

motoneurons to express self-sustained plateau potentials, thereby contributing to muscle spasms (*Nickolls et al., 2004*). In keeping with this hypothesis, chronic spinal rats show the emergence of long-lasting reflexes in vivo (*Murray et al., 2010*), similar to those we recorded in response to Bk in vitro (see *Figure 1D*).

The spinal cord also represents a relevant site of action for kinins to produce hypernociception (*Steranka et al., 1988*; *Perkins et al., 1993*; *Ferreira et al., 2002*). The flexor withdrawal motor reflex has commonly been used as a surrogate pain model (*Barrot, 2012*); however, the contribution of motoneuron hyperexcitability has never been questioned. A role for Bk in hyperalgesic assessment methods at the level of motoneurons remains to be substantiated. It would be expected that excitation of sensory interneurons by Bk works in concert with direct sensitization of motoneurons to augment the withdrawal flexor reflex. It is also important to emphasize that the neonatal response to nociception differs from that found in adults (*Fitzgerald, 2005*). It seems therefore rational that Bk actions might undergo marked postnatal changes.

## Materials and methods

Experiments were performed on neonatal (0–7 days old) Wistar rats of either sex. We housed rodents in a temperature-controlled animal care facility with a 12-hr light–dark cycle and given *ad libitum* access to food and water. We made all effort to minimize animal suffering and the number of animals used. We performed experiments in accordance with French regulations (Ministry of Food, Agriculture and Fisheries; Division of Health and Protection of Animals).

### In vitro preparations

Details of the in vitro preparations have been previously described (*Bouhadfane et al., 2013*; *Brocard et al., 2013*) and are only summarized here. *For the isolated spinal hemicord preparation*, the spinal cord was transected at T10 and longitudinally cut along the midline. The hemispinal cord with the medial side up was transferred to the recording chamber and perfused with oxygenated normal Krebs solution composed of the following (in mM): 120 NaCl, 3 KCl, 1.25 $NaH_2PO_4$, 1.3 $MgSO_4$, 1.2 $CaCl_2$, 25 $NaHCO_3$, 20 D-glucose, pH 7.4, ~32°C pH 7.4. *For the slice preparation*, the lumbar spinal cord was isolated in oxygenated (95% $O_2$/5% $CO_2$) ice-cold (<4°C) low-sodium medium, composed of the following (in mM): 232 sucrose, 3 KCl, 1.25 $KH_2PO_4$, 4 $MgSO_4$, 0.2 $CaCl_2$, 26 $NaHCO_3$, 25 D-glucose, pH 7.4. The lumbar spinal cord was then introduced into a 4% agar solution, quickly cooled, mounted in a vibrating microtome (VT1000S; Leica, Germany) and sliced (350 µm) through the L3-5 lumbar segments. Slices were immediately transferred into the holding chamber filled with oxygenated aCSF solution (composed of [in mM]: 120 NaCl, 3 KCl, 1.25 $NaH_2PO_4$, 1.3 $MgSO_4$, 1.2 $CaCl_2$, 25 $NaHCO_3$, 20 D-glucose, pH 7.4, ~30°C). Following a 1-hr resting period, individual slices were transferred to a recording chamber that was continuously perfused (~4 ml/min) with the same medium heated to ~34°C. All solutions were oxygenated with 95% $O_2$/5% $CO_2$.

### Recordings and stimulation

Electrophysiological data were acquired through a Digidata 1440a interface using the Clampex 10 software (Molecular Devices). *For the spinal hemicord preparation*, motor outputs were recorded using extracellular stainless steel electrodes placed in contact with lumbar ventral roots (L3–L5) and insulated with Vaseline. The ventral root recordings were amplified (×2000), high-pass filtered at 70 Hz, low-pass filtered at 3 kHz, and sampled at 10 kHz. Monopolar stainless steel electrodes

were also placed in contact with the dorsal roots and insulated with Vaseline to deliver a brief supramaximal stimulation (0.2 ms duration). Dorsal root stimulation was repeated five times with an interstimulus interval of 60 s for each trial. *For the slice preparation*, whole-cell patch-clamp recordings were made from motoneurons located in the lateral ventral horn using a Multiclamp 700B amplifier (Molecular Devices, Sunnyvale, CA) in current- or voltage-clamp mode. Motoneurons were visually identified with video microscopy (E600FN; Nikon Eclipse, Japan) coupled to infrared differential interference contrast, as the largest cells located in layer IX. The image was enhanced with a KP-200/201 infrared-sensitive CCD camera (Hitachi, Japan) and displayed on a video monitor. Only one motoneuron was recorded from each slice. Patch electrodes (2–3 MΩ) were pulled from borosilicate glass capillaries (1.5 mm OD, 1.12 mm ID; World Precision Instruments, Sarasota, FL) on a Sutter P-97 puller (Sutter Instruments Company, Novato, CA) and filled with intracellular solution containing (in mM): 140 $K^+$-gluconate, 5 NaCl, 2 $MgCl_2$, 10 HEPES, 0.5 EGTA, 2 ATP, 0.4 GTP, pH 7.3 (280–290 mOsm). In some recordings, BAPTA (10 mM), GDPßS (2 mM), or xestospongin (2.5 mM) were added in the pipette solution to chelate intracellular $Ca^{2+}$, to inhibit G-proteins or to block InsP3 receptors, respectively. Recordings were digitized on-line and filtered at 10 kHz (Digidata 1322A, Molecular Devices). Access resistance was monitored periodically throughout the experiments.

## Two photon imaging

For calcium-indicator loading of cell populations (*Stosiek et al., 2003*), we used multicell bolus loading with Oregon Green 488 BAPTA-1 acetoxymethyl ester (1 mM OGB1-AM; Life Technologies, France), which resulted in the staining of virtually all cells near the ejection site, including neurons and astrocytes. Astrocytes were identified through additional staining with the astrocytic marker Sulforhodamine-101 (300 µM SR-101), which permitted a clear separation of the astroglial (yellow) and neuronal (green) networks. According to the criteria stated above, motoneuron somata were manually selected in a three-dimensional space after acquiring a reference z-stack of the volume on a custom-built three dimensional fast acousto-optical trajectory scanning microscope (Femtonics Ltd, Budapest, Hungary) (*Katona et al., 2012*) with a femtosecond pulsed laser tuned to 830 nm (Mai Tai HP, Spectra Physics, Mountain View, California). For each motoneuron selected, acquisition points were placed in the soma in the cytoplasmic area avoiding the nucleus. The selected neurons were imaged at 33 kHz/point. Imaging was controlled using the MES software package (Femtonics Ltd) based on Matlab (Mathworks). Fluorescence was measured as the average pixel value over selected somatic and proximal dendritic regions. Calcium changes were calculated as the relative change in fluorescence [$\Delta F/F = (F - F_0)/(F_0 - F_b)$], where $F_0$ is baseline fluorescence and $F_b$ is background fluorescence measured from a region lacking labeled cells away from the recorded area.

In some experiments, whole-cell recording and simultaneous calcium imaging were performed on motoneurons identified using 900-nm oblique illumination. In these experiments, the recording electrodes (2–5 MΩ) contained 120 µM OGB-1 and 20 µM Alexa-594 (Life Technologies) dissolved in the same intracellular solution as stated above. To ensure equilibration between the pipette solution and cytosol, the acquisition was started at least 15 min after establishment of the whole-cell configuration. For each subcellular region of interest (soma and dendrites) five to ten acquisition points were manually defined, outside the region of the nucleus, and averaged during analysis. For simultaneous calcium imaging and whole cell recordings, detection of calcium transients was triggered by the onset of spikes or the onset of the Bk response measured in the electrophysiological recording. All imaging sessions were conducted at 34°C.

## Data analysis

Electrophysiological data were analyzed off-line with Clampfit 10 software (Molecular Devices) or the MES sofware package (Femtonics Ltd). Only cells exhibiting a stable holding membrane potential, access resistance (no more than 20% variation) and an action potential amplitude larger than 45 mV were considered. All reported membrane potentials were corrected for liquid junction potentials. Passive membranes properties of cells were measured by determining from the holding potential the largest voltage deflections induced by small current pulses that avoided activation of voltage-sensitive currents. The input resistance and input conductance were measured by the slope of the linear portion of the I/V relationship. There was evidence of inward rectification ('sag') during strong hyperpolarization.

The size of the sag was measured from the voltage response of the cell to the hyperpolarizing current pulse adjusted to elicit peak voltage deflections to −120 mV, and was defined as $100 \times (1 − Vss/Vpeak)$, where Vpeak was the peak voltage deflection from the holding potential (−70 mV) and Vss was the steady-state voltage deflection from the holding potential. Firing properties were investigated with 1-s-long depolarizing current pulses of varying amplitudes. Firing frequency was calculated as the average action potential frequency over the entire duration of the pulse. The rheobase was defined as the minimum current intensity necessary to induce an action potential. To investigate the action potentials and afterhyperpolarizations (AHPs), single spikes were evoked by brief intracellular pulses from a holding potential of −60 mV. Peak spike amplitude was measured from the threshold potential, and spike duration was defined as the width of the action potential at 50% of the peak. The peak amplitude and duration (to half of the peak height) of AHPs were measured from the holding potential. To investigate the sADP, a train of spikes was evoked by a 2-s current pulse at holding potential of −60 mV. The peak amplitude of the sADP was measured from the holding potential. The peak amplitude of the depolarization induced by bath application of Bk was measured from the holding potential. When the Bk-induced depolarization reached a steady-state level, the membrane potential was manually moved back to its original holding level for direct comparison of membrane properties before and during the application of the nanopeptide. A standard sigmoidal curve was fit to the relation between log of agonist dose and the Bk-induced depolarization. The dose that produced 50% of the maximal effect ($EC_{50}$) was measured from the curve. To characterize the ionic mechanism(s) underlying the Bk-evoked depolarization, the membrane potential was initially held in voltage clamp at −70 mV, which approximates the resting membrane potential of motoneurons (*Tazerart et al., 2007*), and then ramped to 0 mV over 3 s. To show the reversal potential, the *I–V* relationship of the current was constructed by subtracting the *I–V* relationship obtained before Bk application (control) from that performed in the presence of the nanopeptide. The current generated was fitted by a linear regression which was used to extrapolate the reversal potential at which the currents intersect.

Motor outputs recorded on the ventral roots in response to a brief dorsal root stimulation were quantified by cumulative counts of spikes generated in PSTHs (bins width: 20 ms) over a time window of 4000 ms post-stimulation. PSTHs were constructed from 5 consecutive rectified responses. As defined previously (*Murray et al., 2010*), we computed the transient short latency and long-lasting reflexes over a window of 10–40 ms and 500–4000 ms post stimulus, respectively. Counts were corrected for spontaneous activity by subtracting the number of spontaneous events arising prior to the stimulus.

Calcium fluorescence changes were analyzed using the MES software package (Femtonics Ltd) based on Matlab (Mathworks).

Data are presented as means ± SEM. The statistical tests used are given in the text and figure legends. Values of $p < 0.05$ were considered significant (GraphPad Software, San Diego California USA).

## Solutions and drug list

Normal aCSF was used in most cases for electrophysiological recordings. In whole-cell current-clamp and voltage-clamp recordings, TTX (0.5–1 µM) was used to eliminate the effect of presynaptic input, except when firing properties of motoneurons were studied. Low $Na^+$ extracellular solution was prepared by replacing NaCl with equivalent concentration of choline chloride. High-$K^+$ aCSF was prepared by adding 1 M KCl. The effect of extracellular pH on membrane current was examined by lowering pH with 1.0 N HCl solution. Drugs were purchased from the following sources: apamin (100 nM), 4-aminopyridine (2 mM), BAPTA (10 mM), barium (100–300 µM), cadmium chloride (100 µM), CTX (2 µg.ml$^{-1}$), chelerythrine (10 µM), CNQX (10 µM), GDPßS (2 mM), [Hyp3]-Bk (2 µM), kynurenate (1.5 mM), pertussis toxin (PTX: 2 µg.ml$^{-1}$), PKA 6–22 (10 µM), PKC 19–36 (10 µM), R 715 TFA (5 µM), tetraethylammonium chloride (TEA, 10 mM) from Sigma–Aldrich; AP-5 (20 µM), bradykinin (Bk, 0.01–16 µM), H89 (10 µM), Hoe 140 (2 µM), KN-93 (2 µM), Lys-[Des-Arg$^9$]-Bk (2 µM), quinidine (200–400 µM), ruthenium red (RR, 10 µM), spantide (2–5 µM), tetrodotoxin citrate (TTX, 20 nM–1 µM), U73122 (10 µM), W-7 (100 µM), ZD7288 (20 µM) from Tocris (Bristol, UK). In most experiments, drugs were delivered via superfusion using a syringe pump. In some experiments, cells were loaded with BAPTA, GDPßS, or xestospongin C by passive diffusion from the patch pipette. In some experiments performed using the in vitro hemicord preparation, a pipette with a wide drip tip containing Bk was placed ~25–50 µM upstream of the ventral horn. Leakage of the solution from the pipette was monitored by including neutral red in the intrapipette solution. This configuration restricts the application of drug to the ventral horn (see *Figure 1A*) and avoids mechanical artifacts.

## Acknowledgements

This work was supported by the French Agence Nationale pour la Recherche (ANR). AK received a grant from the Fondation pour la Recherche Médicale (FRM).

## Additional information

### Competing interests

BR: Founder of Femtonics Ltd and a member of its scientific advisory board. The other authors declare that no competing interests exist.

### Funding

| Funder | Author |
| --- | --- |
| Fondation pour la Recherche Médicale | Attila Kaszás |
| Agence Nationale de la Recherche | Laurent Vinay |

The funders had no role in study design, data collection and interpretation, or the decision to submit the work for publication.

### Author contributions

MB, AK, Acquisition of data, Analysis and interpretation of data, Drafting or revising the article; BR, RMH-W, LV, Analysis and interpretation of data, Drafting or revising the article; FB, Conception and design, Acquisition of data, Analysis and interpretation of data, Drafting or revising the article

### Ethics

Animal experimentation: All animals care and use conformed to the French regulations (Ministry of Food, Agriculture and Fisheries; Division of Health and Protection of Animals; Ministry of Higher Education and Research) and were approved by the local ethics committee (Comité d'Ethique en Neurosciences INT-Marseille, authorization Nb A9 01 13).

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
