## [Decision Letter]

Thank you for sending your work entitled “Sensitization of lumbar motoneurons by the inflammatory pain mediator bradykinin” for consideration at *eLife*. Your article has been favorably evaluated by Eve Marder (Senior editor) and three reviewers, one of whom, Ronald L Calabrese, is a member of our Board of Reviewing Editors.

The Reviewing editor and the other reviewers discussed their comments before we reached this decision, and the Reviewing editor has assembled the following comments to help you prepare a revised submission.

The authors present a thorough pharmacological and electrophysiological analysis of the direct effects of Bradykinin on motoneurons from rat neonatal spinal cord. The data set presented is extensive and the experiments are carefully done. The writing is clear and the figures well designed. Most of the conclusions depend on pharmacological analysis and although the specificity of the drugs is in several cases not high the preponderance of the evidence and the multiple drugs used produce a substantive story. The major conclusion is that Bk acts directly on motor neurons to increase their excitability by activating B2 receptors. The evidence is further interpreted that Bk acts via the single PLC/InsP3/calcium signaling pathway to inhibit leak K^+^ current (likely TWIK channels) and activate ICaN (likely thermo-TRP channels). These results have important implications for enhanced withdrawal reflexes seen in injury-induced hypernociception.

There are several concerns that must be addressed in revision before publication in *eLife*.

1) At several points, the authors speculate that the actions of Bk on motoneurons could influence hyperalgesic assessment methods. While this is not unreasonable, the experiments were all from neonatal rats and it is also possible that hyperalgesic responses and Bk effects on motoneurons might change with age. In addition, all experiments were conducted with the highest concentration of Bk (rather than the EC50). Are these Bk concentrations relevant to in vivo levels? Moreover, there is no evidence presented that stimulation of nociceptive afferents induces bradykinin-dependent sensitization of motoneurons. These limitations should be confronted as the impact of the work on hyperalgesic assessment is described.

2) The authors conclude, from the experiments in Figure 3, that Bk is acting directly on motoneurons. However, it is difficult (if not impossible) to exclude all other possible indirect mechanisms in whole slices. For instance, are Bk receptors expressed on glia? Are glia unresponsive to Bk applications? Given this, the authors should only state that the Bk effect is “likely” via a direct mechanism.

3) This work provides evidence that Bk can act directly on motor output, rather than indirectly via Substance P release. However, the effect of spantide on the Bk response was not shown and there is no positive control for this negative result. It would also be useful to see the effect of spantide in the hemisection experiments.

4) There is no clear demonstration that the effects of leak K^+^ and Na^+^ are two separate effects. Unless this is clearly shown, this conclusion should be toned down.

5) There is a general concern that many of the conclusions are based on the use of compounds that are not entirely specific and act on other conductances. For example, the data implicating thermo-TRP channels are not entirely conclusive (e.g., based on effects of ruthenium red and lowered temperatures, both very non-specific), so that interpretation comes across as too definitive. Moreover the pharmacological implication of TWIK channels is also not iron clad. Perhaps the interpretation should simply be a Bk modulation of leak.

Minor comments:

Reviewer #1:

1) The use of motor circuits in the Abstract, Introduction and Discussion is not appropriate. This study focuses on motoneurons and does not examine the effect of bradykinin on motor circuits.

Reviewer #2:

1) I really want to see the ramp currents and the ramp difference currents over the full voltage range employed. This should be done right up front as the lead in in Figure 4. I also particularly want to see how the Bk-induced ramp currents in the presence of quinidine compared to control Bk-induced currents. Is the I_CaN_ really voltage insensitive? I would also like some discussion of how the two currents affected by Bk give rise to the overall decrease in Rin.

2) I would have liked to see how each of the drugs used in Figure 7 affected the two different current components.

Reviewer #3:

1) It is unclear what the relevance is of comparing calcium responses in the dendrites versus soma? Are intracellular calcium stores primarily located in the soma, not the dendrites (where the more robust calcium response was observed)?

2) Figure 2. The legend describes a 2-s stimulation, which looks like a 5-s in the figure. Also, the spike analysis (threshold, amplitude) is from long depolarizing steps while the AHP analysis is from brief pulses. All that analysis should be based on spikes evoked from brief pulses.

3) Was the sag analysis performed during hyperpolarization to the same voltage under all conditions and from the same voltage? The underlying I_h_ is voltage-dependent and the sag will be different depending on the degree of hyperpolarization.

---

## [Author Response]

*1) At several points, the authors speculate that the actions of Bk on motoneurons could influence hyperalgesic assessment methods. While this is not unreasonable, the experiments were all from neonatal rats and it is also possible that hyperalgesic responses and Bk effects on motoneurons might change with age. In addition, all experiments were conducted with the highest concentration of Bk (rather than the EC50). Are these Bk concentrations relevant to in vivo levels? Moreover, there is no evidence presented that stimulation of nociceptive afferents induces bradykinin-dependent sensitization of motoneurons. These limitations should be confronted as the impact of the work on hyperalgesic assessment is described*.

We agree that the neonatal response is not necessarily a weak form of that found in the adult; rather than a targeted flexion of the paw, the hyperalgesic response in neonatal rats consists of unspecified body movements (Teng and Abott, 1998; Pain, 1998 Jun;76(3):337-47). There is evidence that the postnatal maturation of hyperalgesia arises from fine-tuning of the pain circuitry with age ([32]; Nat Rev Neurosci. 2005 Jul;6(7):507-20.). It seems therefore possible that Bk actions might undergo marked postnatal changes. This potential developmental complication is now mentioned in the last part of the Discussion. The referee is correct that there is no direct evidence demonstrating that the Bk-dependent sensitization of motoneurons influences the withdrawal reflex. We now clearly stipulate that the influence for Bk in hyperalgesic assessment methods at the level of motoneurons remains to be substantiated.

In the framework of our investigation, saturating concentrations were used to activate all the Bk receptors on the motoneurons. Whether the Bk concentrations used have any physiological role is at present unclear; it is likely that, like other peptides, Bk normally acts at lower concentrations.

*2) The authors conclude, from the experiments in*
Figure 3*, that Bk is acting directly on motoneurons. However, it is difficult (if not impossible) to exclude all other possible indirect mechanisms in whole slices. For instance, are Bk receptors expressed on glia? Are glia unresponsive to Bk applications? Given this, the authors should only state that the Bk effect is “likely” via a direct mechanism*.

Given the presence of B2 receptors on motoneurons, and all of the effects of Bk are blocked by B2 antagonists, it is very unlikely that there are no direct effects whatsoever of Bk on these neurons. However, it cannot be completely ruled out that bath-applied Bk activates glial cells, resulting in the release of an unknown chemical substance which acts on motoneurons to elicit additional indirect effects. Microglial cells express functional Bk receptors ([70];Life Sci. 2003 Feb 21;72(14):1573-81; [43]; J Neurosci. 2007 Nov 28;27(48):13065-73) and Bk can act as a neuron-glia signaling molecule ([40]; J Physiol. 2001 Oct 1;536(Pt 1):111-21; [71]; Nature, 1994 Jun 30;369(6483):744-7). The potential cross-link between neuron and glial cells is now clearly described in the Discussion and the references are added (see subsection headed “Functional evidence of a role for K2P and TRP channels”). Furthermore, as requested by the reviewer we soften throughout the manuscript our conclusions on a direct postsynaptic effect of Bk.

*3) This work provides evidence that Bk can act directly on motor output, rather than indirectly via Substance P release. However, the effect of spantide on the Bk response was not shown and there is no positive control for this negative result. It would also be useful to see the effect of spantide in the hemisection experiments*.

The inability of spantide to block the Bk effect is now illustrated in Figure 3—figure supplement 1. As a positive control, we refer to Yanagisawa and Otsuka’s study in Br. J. Pharmacol (1990) where they demonstrate in neonatal rat lumbar motoneurons that spantide, used at the same concentration that we used, significantly depressed the response to substance P but not those to glutamate, noradrenaline and GABA; this is now referenced (see subsection entitled “Bradykinin depolarizes lumbar motoneurons through a likely direct postsynaptic activation”). As requested by the reviewer, we performed additional experiments with spantide in the hemispinal cord preparation. Similar to Bk responses recorded intracellularly from motoneurons, the motor output induced by Bk and recorded extracellularly from ventral roots is not affected by spantide. These new data are now added in the manuscript (see the aforementioned subsection) and illustrated in Figure 3—figure supplement 1.

*4) There is no clear demonstration that the effects of leak K*^*+*^
*and Na*^*+*^
*are two separate effects. Unless this is clearly shown, this conclusion should be toned down*.

There may be a misunderstanding on this point. The effects on the leak K^+^ and Na^+^ currents are blocked by different pharmacological agents. However, we believe that the two ionic mechanisms are synergistic: the depolarizing effect of the Na^+^ current will be enhanced by the simultaneous reduction of a resting K^+^ conductance. This synergistic effect is now clearly explained in the Discussion.

*5) There is a general concern that many of the conclusions are based on the use of compounds that are not entirely specific and act on other conductances. For example, the data implicating thermo-TRP channels are not entirely conclusive (e.g., based on effects of ruthenium red and lowered temperatures, both very non-specific), so that interpretation comes across as too definitive. Moreover the pharmacological implication of TWIK channels is also not iron clad. Perhaps the interpretation should simply be a Bk modulation of leak*.

We agree that, due to the nonspecificity of the available pharmacological agents, it is difficult to determine which type of channels are activated by Bk. To meet the reviewers’ request: 1) We have completely rewritten the Results section by focusing on the nature of conductances without referring to putative channels involved. 2) The discussion about the putative channels involved has been amended to avoid any too definitive conclusion.

Minor comments

Reviewer #1:

*1) The use of motor circuits in the Abstract, Introduction and Discussion is not appropriate. This study focuses on motoneurons and does not examine the effect of bradykinin on motor circuits*.

As requested, “motor circuits” has been changed to “motoneurons”.

Reviewer #2:

*1) I really want to see the ramp currents and the ramp difference currents over the full voltage range employed. This should be done right up front as the lead in in*
Figure 4*. I also particularly want to see how the Bk-induced ramp currents in the presence of quinidine compared to control Bk-induced currents. Is the I*_*CaN*_
*really voltage insensitive? I would also like some discussion of how the two currents affected by Bk give rise to the overall decrease in Rin*.

We have now added the ramp currents with the I/V curves in all relevant figures.

The I_CaN_ investigated in our study is itself voltage insensitive, at least in the voltage range investigated. Whether the current shows an indirect voltage sensitivity depends on whether it is normally activated by calcium entry from voltage-activated calcium channels a, as we previously studied (5). However, this mechanism is not involved in the present study as cadmium block of voltage-activated calcium channels did not affect the Bk-induced depolarization. Our work suggests that voltage-independent release of calcium from internal stores is responsible for activation of I_CaN_ in response to Bk.

The overall increase in input resistance we observe during Bk application reflects the sum of the two currents activated, a conductance decrease to K^+^ and a conductance increase to Na^+^. We were initially surprised to fail to measure the conductance increase to the isolated Na^+^-mediated current. This may reflect a predominantly dendritic location of I_CaN_, where their influence on soma resistance would be difficult to measure. Consistent with this interpretation, our calcium imaging experiments suggest that Bk causes significantly greater rises of Ca^2+^ in the dendrites than the soma. In addition, the B_2_ receptors have been found primarily in the peripheral processes of motoneurons ([60]; Neuroscience. 1995 Oct;68(3):867-81). These points are now discussed (see Discussion section).

*2) I would have liked to see how each of the drugs used in*
Figure 7
*affected the two different current components*.

Such investigation would be interesting to replicate after isolating the two ionic components. However, both currents were mimicked by B2 (and not B1) agonists, and blocked by a B2 antagonist. In addition, most of the drugs used, particularly GDPßS, U73122 and W-7, almost abolish the total response to Bk suggesting that both ionic components are affected. It is likely that B_2_ receptors activate a single intracellular pathway (PLC/Ca^2+^/InsP3/calmodulin) which, in turn, simultaneously modulates the two components.

Reviewer #3:

1) It is unclear what the relevance is of comparing calcium responses in the dendrites versus soma? Are intracellular calcium stores primarily located in the soma, not the dendrites (where the more robust calcium response was observed)?

Intracellular calcium stores are not only found in the soma. The structural and molecular apparatus for Intracellular calcium stores is known to be present in dendrites (Sharp et al. 1993; J Neurosci. 1993 Jul;13(7):3051-63 ; Spacek and Harris, 1997; J Neurosci. 1997 Jan 1;17(1):190-203.). In addition, the contributions of Ca^2+^ release from internal stores to dendritic Ca^2+^ transients following synaptic activation has been previously observed (Alford et al., 1993, J. Physiol. (Lond), 469 (1993), pp. 693–716; Emptage et al., 1999, Neuron. 1999 Jan;22(1):115-24). Comparing calcium responses in the dendrites and the soma gave us important insights. First, the greater increase of [Ca^2+^]_i_ in dendrites compared to the soma suggests that Ca^2+^-activated channels will be more affected in dendrites and thus may act to modify the integration of synaptic inputs of motoneurons. Second, the propagating wave of [Ca^2+^]_i_ initiated at the dendrites before the soma is consistent with the location of B_2_ receptors in the peripheral processes of motoneurons ([60]; Neuroscience. 1995 Oct;68(3):867-81), and suggests that the Bk response arises initially in the dendrites. These points have been added in the Discussion.

*2)*
Figure 2*. The legend describes a 2-s stimulation, which looks like a 5-s in the figure. Also, the spike analysis (threshold, amplitude) is from long depolarizing steps while the AHP analysis is from brief pulses. All that analysis should be based on spikes evoked from brief pulses.*

The scale on Figure 2 has been corrected. As requested, analysis of the spike properties has been performed from a brief depolarization, so all properties are derived from the same measurements. The Methods section has been changed, as well as the values of Results and Table 1.

*3) Was the sag analysis performed during hyperpolarization to the same voltage under all conditions and from the same voltage? The underlying I*_*h*_
*is voltage-dependent and the sag will be different depending on the degree of hyperpolarization.*

Yes, all measurements were made from the same initial hyperpolarization. We have now added in Methods (see subsection headed “Data analysis”) the protocol used to measure the sag: “The size of the sag was measured from the voltage response of the cell to a hyperpolarizing current pulse adjusted to elicit peak voltage deflections to -120 mV, and was defined as 100×(1-Vss/Vpeak), where Vss was the steady-state voltage deflection from the holding potential (-70 mV) and Vpeak was the peak voltage deflection from the holding potential.”